# Random Feature Spiking Neural Networks

## Abstract

Spiking Neural Networks (*SNN*s) as Machine Learning (*ML*) models have recently received a lot of attention as a potentially more energy-efficient alternative to conventional Artificial Neural Networks. The non-differentiability and sparsity of the spiking mechanism can make these models very difficult to train with algorithms based on propagating gradients through the spiking non-linearity. We address this problem by adapting the paradigm of Random Feature Methods (*RFM*s) from Artificial Neural Networks (*ANN*s) to Spike Response Model (*SRM*) *SNN*s. This approach allows training of *SNN*s without approximation of the spike function gradient. Concretely, we propose a novel data-driven, fast, high-performance, and interpretable algorithm for end-to-end training of *SNN*s inspired by the *SWIM* algorithm for *RFM-ANN*s, which we coin *S-SWIM*. We provide a thorough theoretical discussion and supplementary numerical experiments showing that *S-SWIM* can reach high accuracies on time series forecasting as a standalone strategy and serve as an effective initialisation strategy before gradient-based training. Additional ablation studies show that our proposed method performs better than random sampling of network weights.

## 1 Introduction

Ever since their great potential as *ML* models has been shown theoretically (Maass, 1996; 1997b; Maass & Schmitt, 1999), much effort has been devoted to bringing Spiking Neural Networks (Maass, 1997a) to the same degree of maturity as their artificial counterparts. They are regarded as having the potential of being more energy-efficient than conventional artificial neural networks (Yan et al., 2025) when implemented and trained properly. These efficiency gains are mainly enabled by *SNN*s employing sparse computation and communication by reducing the required computation to sparsely distributed short-lived events called spikes. Many works have investigated applying the gradient-based methods, which enabled the massive success of Deep Learning to *SNN*s; however, it is widely accepted that the sparse event-driven nature enabling the high efficiency of *SNN*s, also makes them difficult to train the same way as *ANN*s (Bohte et al., 2002; Neftci et al., 2019; Wu et al., 2018). While recent works have achieved considerable progress in the direction of mitigating the *SNN* specific obstacles to gradient-based training (Hu et al., 2024), these methods remain subject to systematic bias introduced by surrogate-gradient approximations (Gygax & Zenke, 2025) and inadequate utilisation of temporal dynamics (Li et al., 2023), and they inherit the general shortcomings of gradient-based training: Sensitivity to optimisation hyperparameters (Goodfellow et al., 2016), slow convergence rate (Boyd & Vandenberghe, 2004; Bottou et al., 2018), substantial computational and memory costs, as well as a lack of interpretability and an incomplete theoretical characterisation of optimisation landscapes (Taranto & Addie, 2025; Islamov et al., 2024; Mersha et al., 2024). To address these problems, recently, fast and interpretable methods for training *ANN*s based on the *RFM* paradigm, which can train networks orders of magnitude faster than iterative gradient-based methods while achieving competitive accuracies and also being understood much better from a theoretical point of view, have been proposed (Bolager et al., 2023; 2025). *SNN*s have thus far benefited only very little from advances in *RFM*s (Dai & Ma, 2025; Basu et al., 2013). *SNN*s are

meant to be fast and efficient at inference time, but currently, the training time with *SGD* prohibits efficient testing. We aim to mitigate this limitation by translating the modern high-performance *SWIM* (Bolager et al., 2023) algorithm from *ANN*s to *SNN*s, coining the resulting algorithm "*S-SWIM* ".

In short, current approaches for training *SNN*s based on surrogate gradient approximations suffer from being computationally expensive and introducing poorly understood systematic biases. We address this problem by solving simple surrogate objectives for the trainable parameters which do not require approximating or backpropagating spike-function gradients.

In summary, our main contributions include:

**Data-driven Sampling Algorithm for *SNN*s.** To the best of our knowledge, this study is among the first to apply the *RFM* paradigm to *SNN*s and the first ever to propose a data-driven sampling method for joint learning of weights and temporal parameters.

**Theoretical Discussion and Empirical Evaluation.** We provide a thorough theoretical discussion of the proposed method, validated by numerical experiments.

**High Performance Initialisation.** Our numerical experiments show that using the proposed method as an initialisation strategy for gradient-based training leads to higher performance in fewer epochs.

## 2 RELATED WORK

**Regarding *SNN*s for Time Series Forecasting**, it is commonly assumed that *SNN*s are well-suited to tasks such as time series forecasting, by processing information inherently spatiotemporally. Previously, *SNN*s suffered from low performance on these tasks due to the short memory of the commonly employed Integrate-and-Fire (*IF*) (Gerstner & Kistler, 2002, Chapter 1.3) type neuron models (Wang & Yu, 2024). Recent advancements in adapting the neuron models to alleviate this shortcoming by Feng et al. (2025); Lv et al. (2025; 2024) have made *SNN*s efficient and effective on these tasks, providing a solid reference we will compare against in our numerical experiments. **Regarding Delay Learning**, recent works in *SNN* training show a renewed focus on explicit delay-learning: Sun et al. (2023) extend *SLAYER* (Shrestha & Orchard, 2018) by integrating adaptive caps into the delay optimisation and show improved performance on benchmarks with rich temporal dynamics. Hammouamri et al. (2023) introduced an alternative method for parameterising and learning delays by representing them as dilated convolutions with learnable spacings Khalfaoui-Hassani et al. (2023), which can easily be integrated with arbitrary neuron models and gradient-based optimisation pipelines. Deckers et al. (2024) Deckers et al. (2024) demonstrate joint optimisation of weights, delays and neuron model parameters. While these methods highlight the importance of delay learning, they employ gradient-based optimisiation, which is why we instead make use of linear correlation analysis for learning delays to enable fully gradient-free training. **Regarding Random Feature Methods**, while classical methods such as Extreme Learning Machines (*ELM*s) (Huang et al., 2004) can train networks much faster than gradient-based optimisation, they typically require (much) larger networks while performing worse than gradient-trained networks on sufficiently difficult tasks (Gallicchio & Scardapane, 2021). A major shortcoming of *ELM*s is that the weights are chosen entirely data-agnostically, which has been addressed by Bolager et al. (2023) through proposing a data-dependent weight construction and sampling distribution. The main argument is based on explicitly constructing a basis for functions matching the behaviour of the target function, focused on large gradients, thus requiring much fewer neurons than classical *RFM*s to achieve competitive performance, while retaining the speed advantage. Previous work on applying *RFM*s to *SNN*s, such as Basu et al. (2013); Dai & Ma (2025); Wang et al. (2025), used only data-agnostic constructions, which distinguishes our method based on the data-driven *SWIM* method.

## 3 MATHEMATICAL FRAMEWORK

In this section, we introduce our main contribution, the *S-SWIM* algorithm. We start by defining the network architecture and spiking neuron model we want to train, before giving an intuitive overview of the training

algorithm, which we then refine through a technical discussion. See figure 1 for a sketch of the network architecture and the main ideas behind *S-SWIM* . Further details alongside mathematical derivations are provided in section A.

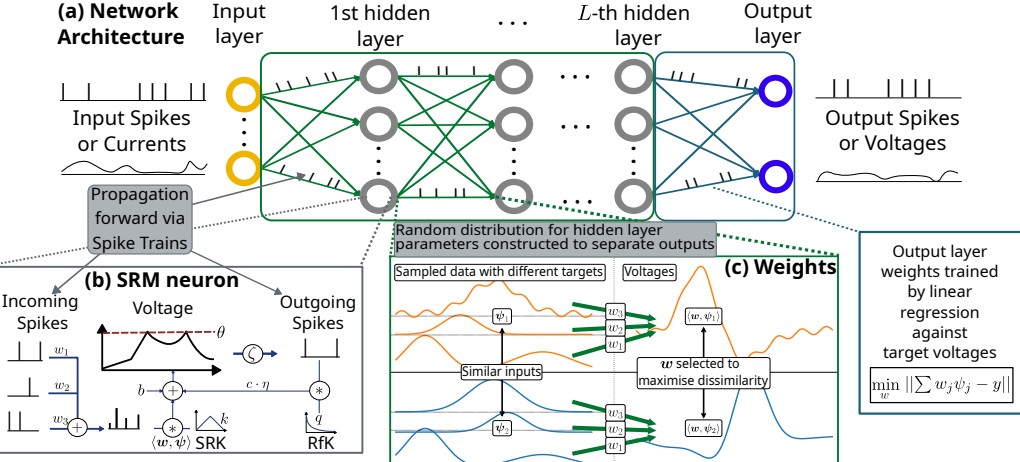

Figure 1: Overview of the network architecture and the main idea of the training algorithm. **(a)** The network consists of successive fully connected layers of *SRM* neurons. Information propagates between layers only as spike trains. Depending on the task, inputs and outputs can be spike trains or real-valued functions. **(b)** Computation of a single *SRM* neuron: Incoming spikes are linearly combined and transformed into a continuous potential through convolution with the spike response kernel (*SRK*) $k$ . After shifting by the bias $b$, outgoing spikes are generated through thresholding. Refractory contributions are generated by convolving the outgoing spikes with the refractory kernel (*RfK*) $q$ and weighting by the spike-cost $c$. Temporal parameters are not shown for simplicity. (cf. definition 3.1). **(c)** The weights of the hidden layers are chosen to separate the membrane potentials generated by samples with similar inputs and dissimilar targets (cf. sections 3.2.2 and 3.2.3). The weights of the final layer are found by solving a linear problem (cf. section 3.2.7).

### 3.1 SPIKE RESPONSE MODEL

While most works on *SNN*s favour *IF*-based neuron models for their easy integrability into existing *ANN* architectures (Ribeiro et al., 2025; Guo et al., 2024; Zou et al., 2025), in this study, we employ the more general *SRM*- which includes the *IF* model by a special choice of kernels (Gerstner et al., 2014, pp. 158–161) - for its mathematical convenience. For a broad discussion of the model, see Gerstner et al. (2014) or Shrestha & Orchard (2018) and Sun et al. (2023) for use as a *ML* model. The version of *S-SWIM* presented here assumes the specific, albeit very flexible, parameterisation of *SRM* based feed-forward spiking neural networks defined in definition 3.1. The overall algorithm is, however, modular, so small adaptations can easily be incorporated into the framework by only changing the respective substep.

**Definition 3.1.** *(Feed-forward Spiking Neural Network) Let* $\mathcal{X} = L^2\left(\mathbb{R}_0^+, \mathbb{R}^{D_{in}}\right)$ *be an input space and* $\mathcal{Y} = L^2\left(\mathbb{R}_0^+, \mathbb{R}^{D_{out}}\right)$ *be an output space. We refer to operators* $\Psi : \mathcal{X} \mapsto \mathcal{Y}$ *with* $\Psi\left(\boldsymbol{x}, t\right) =$

$\left(\Psi_1^{(L+1)}\left(\boldsymbol{x},t\right),\Psi_2^{(L+1)}\left(\boldsymbol{x},t\right),\ldots,\Psi_{D_{out}}^{(L+1)}\left(\boldsymbol{x},t\right)\right)^{\top}$ *of the form*

$$
\Psi_i^{(l)}\left(\boldsymbol{x},t\right)=\begin{cases} x_i(t), & \text{for } l=0 \\ \zeta\left(\sum_{j=1}^{N_{l-1}} W_{ij}^{(l)}\psi_{ij}^{(l)}\left(\boldsymbol{x},\cdot\right)+c_i^{(l)}\eta_i^{(l)}\left(\boldsymbol{x},\cdot\right)+b_i^{(l)},t\right), & \text{for } 0<l\leq L \\ \sum_{j=1}^{N_L} W_{ij}^{(L+1)}\psi_{ij}^{(L+1)}\left(\boldsymbol{x},\cdot\right)+b_i^{(L+1)}, & \text{for } l=L+1 \end{cases} \tag{1}
$$

*with $i\in\{1,2,\ldots N_l\}$ for each $l$ as Feed-forward Spiking Neural Networks (FF-SNNs) with L hidden layers, where $\psi$ denotes the contributions to the membrane potential from the previous layer and $\eta$ the refractory contributions from outgoing spikes, respectively defined as*

$$
\psi_{ij}^{(l)}\left(\boldsymbol{x},t\right)=\left[\left.k^{(l)}\left(\frac{\cdot-\tau_i^{(l)}}{\sigma_i^{(l)}}\right)\right|_{\cdot\geq 0}*\Psi_j^{(l-1)}\left(\boldsymbol{x},\cdot\right)\right](t), \tag{2}
$$

$$
\eta_i^{(l)}\left(\boldsymbol{x},t\right)=\left[\left.q^{(l)}\left(\frac{\cdot}{\varsigma_i^{(l)}}\right)\right|_{\cdot>0}*\Psi_i^{(l)}\left(\boldsymbol{x},\cdot\right)\right](t)=\sum_{t^f\in\mathbb{T}_i^{(l)}(t)}q^{(l)}\left(\frac{t-t^f}{\varsigma_i^{(l)}}\right). \tag{3}
$$

*Here, $[f*g]$ denotes **convolution** and the **Spike-Response-Kernel** (SRK) $k^{(l)}$ and **Refractory-Kernel** (RfK) $q^{(l)}$ are arbitrary $L^2\left(\mathbb{R},\mathbb{R}\right)$ functions defining the temporal shape of input and refractory contributions to the membrane potential of neurons in layer $l$. $\left.f(t)\right|_{t>0}=f(t)$ if $t>0$ and $0$ otherwise (analogously for $\geq$) denotes **half-wave rectification** to avoid dependence on future time points. $\zeta(v,t)=\sum_{t^f\in\mathbb{T}(t)}\delta(t-t^f)$ with the past firing times $\mathbb{T}(t)=\{t^f|v(t^f)\geq 1\wedge t^f\leq t\}$, respectively defined for each neuron, is the **thresholding operator** mapping voltages onto spike trains. $\{\boldsymbol{W}^{(l)},\boldsymbol{b}^{(l)},\boldsymbol{\tau}^{(l)},\boldsymbol{\sigma}^{(l)}\}_{l=1}^{L+1}\cup\{\boldsymbol{c}^{(l)},\boldsymbol{\varsigma}^{(l)}\}_{l=1}^{L}$ are the **trainable parameters** of $\Psi$, referred to as Weights, Biases, (SRK-)Delays, SRK-Supports, Spike-Costs, and RfK-Supports respectively, where $\boldsymbol{W}^{(l)}\in\mathbb{R}^{N_l\times N_{l-1}}$, and $\boldsymbol{b}^{(l)},\boldsymbol{\tau}^{(l)},\boldsymbol{\sigma}^{(l)},\boldsymbol{c}^{(l)},\boldsymbol{\varsigma}^{(l)}\in\mathbb{R}^{N_l}$. Delays and supports together form the temporal parameters of the network. $N_l$ is the number of neurons in the $l$-th layer with $N_0=D_{in}$ and $N_{L+1}=D_{out}$.*

**Remark 3.2.** *definition 3.1 defines FF-SNNs as they are used for regression tasks with real-valued inputs and outputs. For the case of spike-valued inputs, the model stays the same, but the input space $\mathcal{X}$ instead consists of spike-trains. For spike-valued outputs, the last layer $(L+1)$ is defined in the same way as the interior $(0<l\leq L)$ layers, and the output space $\mathcal{Y}$ consists of spike-trains.*

### 3.2 Random Feature Spiking Neural Networks

The key idea behind S-SWIM is to replace iterative gradient-based optimisation of the network parameters with drawing them from a probability distribution. We construct this distribution such that the resulting neurons maximally separate data points with very dissimilar target values. The sampled weights are rescaled to keep the membrane potentials in a reasonable range with respect to the spike threshold. The temporal parameters in the hidden layers are selected heuristically to integrate information from a diverse set of time-scales. Finally, the last layer's delays are found through correlation analysis, while the weights are solved for by constructing an appropriate least-squares problem. In the following, we will make these notions mathematically precise.

The overall structure closely follows the *SWIM* algorithm (Bolager et al., 2023, Algorithm 1). The main differences in the overall outline are that, in addition to weights and biases, temporal parameters have to be specified, and a more involved weight construction is needed due to the data being spatio-temporal. *S-SWIM* does not depend on or assume a specific discretisation in time, so we will use the continuous version in the following. An outline of the major steps of the algorithm is presented in algorithm 1. The individual substeps will be discussed in the following paragraphs, focusing on a compact representation

of the algorithm, such that it could be implemented, with a brief motivation, while deferring the detailed derivations to the Appendix.

### 3.2.1 Temporal Parameters

We will start by defining the substep $\mathcal{T}$, which assigns the temporal parameters in the hidden layers. The output layer requires an interval of length $H$ within $I^{\text{tot}} = [0, T + H)$ containing spikes to fit the target functions $Y$, which are supported on $[T, T + H)$. To ensure the availability of past and recent information, we use the following heuristic in *S-SWIM* . For the neurons in layer $l$, we assign the delays $\boldsymbol{\tau}^{(l)}$ linearly spaced over $\left[0, \frac{l}{L}\tau_{\max}\right)$, with appropriately chosen $\tau_{\max} \in \mathbb{R}_0^+$, so both past and recent information is available to the current layer. If $O < H$, $\tau_{\max}$ should be at least $H - O$ to guarantee a sufficiently long interval of spikes. For the *SRK*-supports $\boldsymbol{\sigma}^{(l)}$, we want multiple scales to be available across different delays, so we cycle through a set of small to large values, defined by a (small) minimum and (large w.r.t. $O$) maximum value $\sigma_{\min}^{\text{h}}, \sigma_{\max}^{\text{h}} \in \mathbb{R}^+$, and a cycle length $N_\sigma^{\text{h}} \ll N_l \in \mathbb{N}$. We then assign the supports linearly spaced between the bounds, repeated every $N_\sigma^{\text{h}}$ neurons. Since the *RfK*s mainly serve to bound the spiking rate and the main arguments of section 3.2.3 and section 3.2.4 focus on the behaviour of the *SRK* contributions, we set $\varsigma_i^{(l)} = \sigma_{\min}^{\text{h}}$ for all neurons, leaving more involved considerations to future work. The temoral parameters are thus given by $\mathcal{T}\left(l, L, N_l, O, H\right) =$

$$\left( \left( \frac{i-1}{N_l} \cdot \frac{l \cdot \tau_{\max}}{L} \right)_{i=1}^{N_l}, \quad \left( \frac{(i-1) \bmod N_\sigma^{\text{h}}}{N_\sigma^{\text{h}} - 1}(\sigma_{\max}^{\text{h}} - \sigma_{\min}^{\text{h}}) + \sigma_{\min}^{\text{h}} \right)_{i=1}^{N_l}, \quad \left( \sigma_{\min}^{\text{h}} \right)_{i=1}^{N_l} \right). \tag{4}$$

### 3.2.2 Sampling Distribution

Next, we will discuss the distribution from which pairs of input signals $x^{(n)}(t)$ are sampled before giving the construction for the associated weights. As in *SWIM*, we sample according to a notion of gradients, however, the gradients are computed in spaces of multivariate functions in our case. The distribution depends on the functions $d_{\tilde{\mathcal{X}}_0}(\cdot, \cdot), d_{\tilde{\mathcal{X}}_l}(\cdot, \cdot), d_{\tilde{\mathcal{Y}}}(\cdot, \cdot)$, defining a notion of distance on the respective spaces. Notably, the inputs and outputs to the network could be real- or spike-valued, while the inputs to interior ($l > 1$) hidden layers are spike-valued, so distances need to be defined for both cases. The choice of distance functions is important, as the weights of the hidden layers are constructed so that they separate (cf. section 3.2.3) samples with similar inputs and dissimilar outputs with respect to the employed notion of distance. In other words, the distance functions need to reflect what features of the data are important, since the hidden layers are constructed to "emphasise" differences in those features. While there is some freedom in choosing suitable functions for the task, they need to fulfill symmetry, definiteness ($d(x, x) = 0$), non-negativity and boundedness, so $P^{(l)}$ can be normalised to a meaningful probability. A possible framework a more formal charactersiation of possible distance functions for both the spiking and real-valued cases is given in section A.1.

### 3.2.3 Sampled weights

Next, we will discuss possible weight constructions $\mathcal{W}$, i.e. how the weights of a given neuron $i$ in layer $l$, $W_{i,:}^{(l)}$, are chosen once the temporal parameters of the given layer have been initialised and input pairs for each neuron have been sampled. Recalling that the input pairs were sampled as having dissimilar outputs, it is natural to require the hidden layer to separate those samples in its output. As in *SWIM*, we split the problem into first finding meaningful directions for the weights before rescaling them and applying the bias to achieve a meaningful output. In *S-SWIM*, this means first finding weights that produce voltage traces ensuring the separation, which is the content of this section, and then scaling and setting the remaining parameters in the current layer to generate reasonable spike trains from those voltage traces, which will be discussed further below. Formally, this separation can be achieved by maximising the distance ("dist") or

---

**Algorithm 1** The *S-SWIM* algorithm. $\mathcal{L}$ is the loss function, which in our case is always the $L^2$ loss. $\epsilon$ serves to bound the sampling distribution and was always set to $\epsilon = 10^{-6}$ in our case. $O, T, H$ depend on the specific task; for time series forecasting $T = O$, and $H$ is the prediction length. $\Psi$ is a *FF-SNN* with fixed architecture that is constructed step-wise according to the algorithm by setting its parameters. $\Gamma$ is the unknown ground-truth mapping to be approximated. The inputs can be real or spike-valued, the outputs must be real-valued (either given from the task or generated surrogate voltages). Line comments indicate where details about the specific substeps can be found.

---

**Constants:** $\epsilon \in \mathbb{R}_{>0}, \; L \in \mathbb{N}_{>0}, \; \{N_l \in \mathbb{N}_{>0}\}_{l=1}^{L+1}$      ▷ *Fixed or part of the model architecture*

**Algorithm Parameters:**      ▷ *Specific to S-SWIM*

     Temporal Parameters $\mathcal{T}$,

     Distances $d_{\mathcal{X}_0}(\cdot, \cdot), d_{\mathcal{X}_l}(\cdot, \cdot), d_{\mathcal{Y}}(\cdot, \cdot)$,

     Weight Construction Criterion $\mathcal{W}$,

     Normaliser $\mathcal{Z}$,

     Initialisation batch size: $\tilde{M} \in \mathbb{N}_{>0}$

**Data:** $X = \{\boldsymbol{x}^{(n)}(t) : \boldsymbol{x}^{(n)} \in \mathcal{X}, n = 1, 2, \ldots, M; t \in [0, O]\}$,

     $Y = \{\boldsymbol{y}^{(n)}(t) : \Gamma\{\boldsymbol{x}^{(n)}\}(t) = \boldsymbol{y}^{(n)}(t) \in \mathbb{R}^{D_{\text{out}}}, n = 1, 2, \ldots, M; t \in [T, T+H]\}$

1:   $X_I = \{\boldsymbol{x}_I^{(n)}\}_{n=1}^{\tilde{M}} \sim \text{Uniform}(X), Y_I = \{\boldsymbol{y}_I^{(n)} : \Gamma\{\boldsymbol{x}_I^{(n)}\} = \boldsymbol{y}_I^{(n)}\}_{n=1}^{\tilde{M}}$

     ▷ *Hidden Layers*

2: **for** $l = 1, 2, \ldots, L$ **do**

     ▷ *section 3.2.1 (Temporal Parameters)*

3:     Set $\Psi : \boldsymbol{\tau}^{(l)}, \boldsymbol{\sigma}^{(l)}, \boldsymbol{\varsigma}^{(l)} \leftarrow \mathcal{T}(l, L, N_l, O, H)$

     ▷ *section 3.2.2 (Sampling Distribution)*

4:     $K \leftarrow \frac{\tilde{M}^2 - \tilde{M}}{2}$      ▷ *Lower triangular part of distance matrix (symmetry)*

5:     $P^{(l)} \in \mathbb{R}^K, \quad P_i^{(l)} \leftarrow 0 \; \forall i$

6:     **for** $n = 1, 2, \ldots, \tilde{M} - 1$ **do**

7:        **for** $m = 1, 2, \ldots, n - 1$ **do**      ▷ *Output of previous layer defines new input space*

8:          $\tilde{\boldsymbol{x}}_I^{(n)}, \tilde{\boldsymbol{x}}_I^{(m)} \leftarrow \Psi^{(l-1)}\left(\boldsymbol{x}_I^{(n)}, [0, T+H]\right), \; \Psi^{(l-1)}\left(\boldsymbol{x}_I^{(m)}, [0, T+H]\right)$

9:          $P_{\text{flat}(n,m)}^{(l)} \leftarrow \dfrac{d_{\tilde{\mathcal{Y}}}\left(\boldsymbol{y}^{(n)}, \boldsymbol{y}^{(m)}\right)}{d_{\tilde{\mathcal{X}}_{l-1}}\left(\tilde{\boldsymbol{x}}_I^{(n)}, \tilde{\boldsymbol{x}}_I^{(m)}\right) + \epsilon}$      ▷ *Indexing of $P^{(l)}$ is arbitrary*

     ▷ *section 3.2.3 (Sampled weights)*

10:     **for** $i = 1, 2, \ldots, N_l$ **do**

11:        Sample $\boldsymbol{x}_I^{(n)}, \boldsymbol{x}_I^{(m)}$ from $X_I$ without replacement, probability $\propto P^{(l)}$

12:        $\boldsymbol{\psi}_1, \boldsymbol{\psi}_2 \leftarrow \psi_{i,:}^{(l)}\left(\boldsymbol{x}_I^{(n)}, [0, T+H]\right), \; \psi_{i,:}^{(l)}\left(\boldsymbol{x}_I^{(m)}, [0, T+H]\right)$      ▷ *Apply SRK*

13:        $\tilde{\boldsymbol{w}} \leftarrow \mathcal{W}(\boldsymbol{\psi}_1, \boldsymbol{\psi}_2)$      ▷ *Candidate weight directions from separation criterion*

     ▷ *section 3.2.4 (Normalisation)*

14:        $\alpha, \beta, \gamma \leftarrow \mathcal{Z}\left(\tilde{\boldsymbol{w}}, \psi_{i,:}^{(l)}(X_I, [0, T+H])\right)$

15:        Set $\Psi : W_{i,:}^{(l)}, b_i^{(l)}, c_i^{(l)} \leftarrow \alpha\tilde{\boldsymbol{w}}, \beta, \frac{\gamma}{q^{(l)}(0)}$      ▷ *Rescale to control voltage statistics*

     ▷ *Output Layer*

     ▷ *section 3.2.5 (Output Delays)*

16: Set $\Psi : \tau^{(L+1)} \leftarrow \mathcal{C}\left(\Psi^{(L)}(X_I, [0, T+H]), Y_I\right)$

     ▷ *section 3.2.6 (Kernel Supports)*

17: Set $\Psi : \sigma^{(L+1)} \leftarrow \Sigma\left(\Psi^{(L)}(X_I, [T, T+H]), Y_I\right)$

     ▷ *section 3.2.7 (Output Weights)*

18: Set $\Psi : \boldsymbol{W}^{(L+1)}, \boldsymbol{b}^{(L+1)} \leftarrow \arg\min_{\boldsymbol{W}, \boldsymbol{b}} \mathcal{L}(\Psi^{(L)}(X, [T, T+H]), Y)$

19: **return** $\Psi$

minimising the inner products ("dot") between the contributions to the voltage associated to the sampled inputs in function space. Those objectives respectively lead to the constructions

$$
\mathcal{W}^{\text{dist}}\left(\boldsymbol{\psi}_1, \boldsymbol{\psi}_2\right) = w_{\max}\left\{\int_0^{T+H} \left(\boldsymbol{\psi}_1(t) - \boldsymbol{\psi}_2(t)\right)\left(\boldsymbol{\psi}_1(t) - \boldsymbol{\psi}_2(t)\right)^\top \mathrm{d}t\right\},
$$

$$
\mathcal{W}^{\text{dot}}\left(\boldsymbol{\psi}_1, \boldsymbol{\psi}_2\right) = w_{\min}\left\{\frac{1}{2}\int_0^{T+H} \boldsymbol{\psi}_1(t)\boldsymbol{\psi}_2(t)^\top + \boldsymbol{\psi}_2(t)\boldsymbol{\psi}_1(t)^\top \mathrm{d}t\right\},
$$

(5)

where $w_{\max}\{\boldsymbol{A}\}$ and $w_{\min}\{\boldsymbol{A}\}$ respectively denote the eigenvectors to the algebraically largest and smallest eigenvalue of $\boldsymbol{A}$. A formal derivation and an intuitive explanation of the proposed weights is given in section A.2.

### 3.2.4 NORMALISATION

The final step $\mathcal{Z}$ in the initialisation of each hidden layers is normalising the voltages to produce "reasonable" spike trains by re-scaling the sampled weights and choosing a suitable bias. Informally, to propagate usable information to the following layers, a spike train should contain some spikes but not too many. For this, we adapt the idea of Rossbroich et al. (2022) and enforce contraints on the statistics of the input contributions to the voltage $\psi_{i,n}^{(l)}$, specifically, the (empirical) expected temporal mean $\mathbb{E}_{v_i}^X$ and standard deviation $\mathbb{S}_{v_i}^X$ approximated over the subbatch $X_I$. One possible constraint is to explicitly prescribe the target mean $\mathbb{E}_{v_i}^{X_I} \overset{!}{=} \mu_t$ and standard deviation $\mathbb{S}_{v_i}^{X_I} \overset{!}{=} s_t$ ("MS"), where $\mu_t \in (-\infty, 1)$ and $s_t \in \mathbb{R}^+$ are tunable hyperparameters. Alternatively, we can prescribe a degree of fluctuation ("FL") by leaving the standard deviation as is and setting the bias to put the mean $r$ standard deviations away from the threshold $\theta = 1$ by $\mathbb{E}_{v_i}^{X_I} \overset{!}{=} 1 - r\mathbb{S}_{v_i}^{X_I}$ with $r \in \mathbb{R}^+$ as tunable parameter. Finally, to make spiking multiple times in quick succession only possible when inputs are especially well aligned with the neurons weights, a sensible choice for the spike cost is $c_i^{(l)} = -3\,\mathbb{S}_{v_i}^{X_I}/q^{(l)}(0)$. These constraints are enforced by relating them to the bias and scale of the weights, as shown in section A.3. This leas to the normalisers $\mathcal{Z}_{\mu_t, s_t}^{\text{MS}}\left(\tilde{\boldsymbol{w}}, \left\{\boldsymbol{\psi}_{i,n}^{(l)}\right\}_{n=1}^{\tilde{M}}\right) =$

$$
\left(\frac{s_t}{\mathbb{E}_n^{X_I}\left[\sqrt{\text{Var}_t^{X_I}\left[\left\langle\tilde{\boldsymbol{w}}, \boldsymbol{\psi}_{i,n}^{(l)}(t)\right\rangle_{\ell^2}\right]}\right]}, \quad \mu_t - s_t\frac{\mathbb{E}_n^{X_I}\left[\mathbb{E}_t^{X_I}\left[\left\langle\tilde{\boldsymbol{w}}, \boldsymbol{\psi}_{i,n}^{(l)}(t)\right\rangle_{\ell^2}\right]\right]}{\mathbb{E}_n^{X_I}\left[\sqrt{\text{Var}_t^{X_I}\left[\left\langle\tilde{\boldsymbol{w}}, \boldsymbol{\psi}_{i,n}^{(l)}(t)\right\rangle_{\ell^2}\right]}\right]} + \Delta^{\text{SC}}, \quad -3\cdot s_t\right)
$$

(6)

and $\mathcal{Z}_r^{\text{FL}}\left(\tilde{\boldsymbol{w}}, \left\{\boldsymbol{\psi}_{i,n}^{(l)}\right\}_{n=1}^{\tilde{M}}\right) =$

$$
\left(1, \quad 1 - r\mathbb{E}_n^{X_I}\left[\sqrt{\text{Var}_t^{X_I}\left[\left\langle\tilde{\boldsymbol{w}}, \boldsymbol{\psi}_{i,n}^{(l)}(t)\right\rangle_{\ell^2}\right]}\right] - \mathbb{E}_n^{X_I}\left[\mathbb{E}_t^{X_I}\left[\left\langle\tilde{\boldsymbol{w}}, \boldsymbol{\psi}_{i,n}^{(l)}(t)\right\rangle_{\ell^2}\right]\right] + \Delta^{\text{SC}}, \quad -3\cdot\mathbb{S}_{v_i}^{X_I}\right).
$$

(7)

The "Silence Correction" ("SC") term $\Delta^{\text{SC}}$ is chosen just large enough to guarantee that each neuron spikes at least once. Detailed derivations for the given expressions and how to choose $\Delta^{\text{SC}}$ can be found in section A.3.

### 3.2.5 OUTPUT DELAYS

After the hidden layers have been initialised, that is all parameters in $\Psi^{(L)}$ have been specified, the remaining parameters $\boldsymbol{W}^{(L+1)}, \boldsymbol{b}^{(L+1)}, \boldsymbol{\tau}^{(L+1)}$, and $\boldsymbol{\sigma}^{(L+1)}$ have to be chosen such that the output of the network best approximates the targets $Y$, which we assume to be real-valued functions throughout the following

discussion. See section A.4 for a discussion of the spike-valued case. The delays $\boldsymbol{\tau}^{(L+1)}$ can be found through linear correlation analysis. Concretely, we set $\mathcal{C}\left(\left\{\Psi^{(L)}\left(\boldsymbol{x}_I^{(n)}\right)\right\}_{n=1}^{\tilde{M}}, \left\{\boldsymbol{y}_I^{(n)}\right\}_{n=1}^{\tilde{M}}\right) =$

$$\left(\arg\max_{\tau \in [0,O)} \sum_{n=1}^{\tilde{M}} \sum_{j=1}^{N_L} \left|\left[\Psi_j^{(L)}\left(\boldsymbol{x}_I^{(n)}\right) \star \left(y_i^{(n)} - \mu_{y_i^{(n)}}(T, T+H)\right)\right]\right| - \Delta^k\right)_{i=1}^{N_{L+1}}, \tag{8}$$

where $[f \star g]$ denotes the cross-correlation of $f$ and $g$ and $\Delta^k$ is the location where the *SRK* peaks. This criterion is quite intuitive: We select the delay at which the incoming spike trains best predict the targets. It can also be derived from explicitly prescribing the definition of $\Psi^{(L+1)}$ as ansatz for $Y$, as is done in section A.5.

### 3.2.6 KERNEL SUPPORTS

Next, the kernel supports $\boldsymbol{\sigma}^{(L+1)}$ need to be specified. *S-SWIM* reduces this to a one-dimensional optimisation problem by aggregating the delays, leading to a shared system matrix for all neurons, and a simplification of the remaining linear problem enabling a fast search over well-chosen candidate values. Concretely, for each candidate, we only need to evaluate the remaining residual norm $\boldsymbol{r}_{i|\sigma_c}^*$ over the subbatch $X_I$ when using the optimal weights and bias, but we do not need to compute the actual parameters. Thus, we define a set of candidates $P$ and for each neuron select

$$\Sigma_P = \left(\arg\min_{\sigma_c \in P} \left\|\boldsymbol{r}_{i|\sigma_c}^*\right\|_2^2\right)_{i=1}^{N_{L+1}} = \left(\arg\min_{\sigma_c \in P} \|\mathcal{D}\boldsymbol{y}_i\|_2^2 - \left\|\boldsymbol{Q}_{\sigma^c}^\top \mathcal{D}\boldsymbol{y}_i\right\|_2^2\right)_{i=1}^{N_{L+1}}, \tag{9}$$

where $\boldsymbol{Q}_{\sigma^c}$ comes from the thin *QR*-factorisation of the augmented design matrix $\boldsymbol{A}(\sigma^c) \in \mathbb{R}^{(H_D \cdot \tilde{M}) \times (N_L+1)}$ built from the discretised *SRK* contributions, computed with the aggregated delay and the candidate $\sigma_c$, as columns concatenated along the temporal dimension, and $\mathcal{D}\boldsymbol{y}_i \in \mathbb{R}^{H_D \cdot \tilde{M}}$ denotes an arbitrary discretisation of the analogously stacked targets $\boldsymbol{y}_i^{(n)}$ for neuron $i$. For the detailed derivations and how the candidates in *S-SWIM* are selected, we refer to section A.6.

### 3.2.7 OUTPUT WEIGHTS

The final step of the algorithm is to solve a linear problem for the output weights $\boldsymbol{W}^{(L+1)}$ and biases $\boldsymbol{b}^{(L+1)}$. Due to the added temporal dimension, the approach applied in the previous section over the subbatch $X_I$ can not be applied to the whole training set. The memory requirements and computational cost grow prohibitely large for any matrix factorisation based approach scaling as $\mathcal{O}\left((M \cdot H_D)^3\right)$ directly applied to equation 68, especially since the system has to be solved independently for each neuron due to the temporal parameters. To avoid these problems, we use a generalised form of the Normal Equations derived in section A.7, which can be assembled in batches and only requires a small linear system scaling only with the number of parameters per neuron to be solved. Concretely, we reformulate the problem as

$$\mathbb{R}^{N_L+1 \times N_L+1} \ni \sum_{n=1}^M \mathcal{D}\boldsymbol{F}_i^{(n)\top} \mathcal{D}\boldsymbol{F}_i^{(n)} + M\lambda \mathbf{Id}_{N_L+1} = \mathcal{D}\boldsymbol{F}_i^{(n)\top} \mathcal{D}\boldsymbol{y}_i^{(n)} \in \mathbb{R}^{N_L+1}, \tag{10}$$

where $\mathcal{D}\boldsymbol{F}_i^{(n)} \in \mathbb{R}^{H_D \times N_L+1}$ is the augmented design matrix for neuron $i$ containing the discretised *SRK* contributions $\boldsymbol{\psi}_{i:}^{(L+1)}$ and a column of 1s for the bias, and $\mathcal{D}\boldsymbol{y}_i^{(n)} \in \mathbb{R}^{H_D}$ are the discretised targets for neuron $i$. This formulation additionally allows for a cheap search over the regularisation parameter $\lambda$, which, alongside the conditioning of the resulting system matrix, is discussed in section A.7.

# 4 NUMERICAL EXPERIMENTS

Table 1: Experimental results of time-series forecasting on 4 benchmarks with various prediction lengths 6, 24, 48, 96. The reference values are sourced from Lv et al. (2025)Table 1. Results highlighted with shading are ours. "Kernel" denotes the *SRK* used in the first layer. Bold font indicates the best *SNN* result. Underlined results indicate *S-SWIM* performing at least as well as gradient-based methods. Italic font indicates that *SGD* optimisation did not fully converge, i.e. the best epoch was within 30 epochs of the maximum. Results are given in the *RSE* Metric, where lower is better. All results are averaged across 3 seeds.

| Models | Comment | | Metr-la ($L=12$) | | | | Pems-bay ($L=12$) | | | | Solar ($L=168$) | | | | Electricity ($L=168$) | | | | Avg. |
| --- | --- | --- | --- | --- | --- | --- | --- | --- | --- | --- | --- | --- | --- | --- | --- | --- | --- | --- | --- |
| | Spike | Kernel | 6 | 24 | 48 | 96 | 6 | 24 | 48 | 96 | 6 | 24 | 48 | 96 | 6 | 24 | 48 | 96 | |
| Transformer w/ RoPE | ✗ | – | .548 | .696 | .802 | .878 | .499 | .563 | .600 | .617 | .225 | .373 | .492 | .539 | .251 | .274 | .341 | .420 | .507 |
| Transformer w/ Sin-PE | ✗ | – | .551 | .704 | .808 | .895 | .502 | .558 | .610 | .618 | .223 | .377 | .504 | .545 | .260 | .277 | .347 | .425 | .513 |
| Spikformer w/ RoPE | ✗ | – | .584 | .757 | .835 | .920 | .519 | .591 | .614 | .625 | .294 | .441 | .550 | .633 | .375 | .383 | .384 | .454 | .560 |
| Spikformer w/ CPG-PE | ✓ | – | .553 | .720 | .806 | .890 | .508 | .580 | .602 | .622 | .257 | .420 | .506 | .555 | .299 | .310 | .314 | .355 | .519 |
| Spikformer-XNOR w/ Conv-PE | ✓ | – | .559 | .721 | .813 | .910 | .518 | .599 | .613 | .628 | .273 | .421 | .527 | .595 | .365 | .371 | .376 | .384 | .542 |
| Spikformer-XNOR w/ Gray-PE | ✓ | – | .546 | .706 | .806 | .885 | .506 | .578 | .597 | .618 | .257 | .409 | .507 | .546 | .276 | .304 | .320 | .342 | .513 |
| Spikformer-XNOR w/ Log-PE | ✓ | – | .543 | .719 | .799 | .876 | .496 | .575 | .601 | .620 | .265 | .408 | .504 | .525 | .272 | .300 | .314 | .340 | .510 |
| SDT-V1 w/ CPG-PE | ✓ | – | .585 | .724 | .799 | .920 | .515 | .578 | .633 | .642 | .285 | .439 | .558 | .637 | .361 | .368 | .370 | .376 | .549 |
| SDT-V1 w/ Log-PE | ✓ | – | .554 | .713 | .807 | .904 | .502 | .585 | .629 | .641 | .280 | .437 | .527 | .598 | .353 | .356 | .360 | .366 | .538 |
| QKFormer w/ Conv-PE (Original) | ✓ | – | .561 | .735 | .832 | .917 | .521 | .586 | .609 | .635 | .289 | .515 | .716 | .784 | .306 | .319 | .355 | .367 | .565 |
| QKFormer w/ CPG-PE | ✓ | – | .536 | .704 | .803 | .896 | .503 | .578 | .589 | .633 | .285 | .520 | .581 | .645 | .266 | .312 | .315 | .332 | .531 |
| QKFormer-XNOR w/ Gray-PE | ✓ | – | .534 | .711 | .804 | .898 | .484 | .577 | .601 | .616 | .276 | .438 | .556 | .570 | .277 | .310 | .314 | .331 | .519 |
| QKFormer-XNOR w/ Log-PE | ✓ | – | .535 | .715 | .805 | .903 | .482 | .581 | .585 | .629 | .274 | .437 | .515 | .564 | **.264** | **.285** | **.296** | .328 | .512 |
| SGD (Morlet) | ✓ | Morlet | .429 | .538 | .615 | .708 | .413 | **.422** | .544 | .804 | **.221** | .357 | .378 | .376 | .302 | .343 | .379 | .570 | .463 |
| SGD (Hat) | ✓ | Hat | .432 | **.528** | **.612** | **.677** | .426 | .455 | .545 | .770 | .256 | .387 | .392 | .371 | .345 | .358 | .431 | .609 | .475 |
| SSWIM (Morlet) | ✓ | Morlet | .507 | .651 | .729 | .806 | .416 | .514 | .589 | .651 | .428 | .371 | .367 | .370 | .378 | .363 | .364 | .370 | .492 |
| SSWIM (Hat) | ✓ | Hat | .513 | .637 | .718 | .794 | .420 | .512 | .583 | .652 | .387 | .369 | .347 | .370 | .429 | .433 | .437 | .433 | .502 |
| SSWIM+SGD (Morlet) | ✓ | Morlet | .464 | .629 | .697 | .736 | **.390** | .468 | .550 | **.602** | .360 | .298 | .279 | .267 | .325 | .313 | .307 | **.308** | .437 |
| SSWIM+SGD (Hat) | ✓ | Hat | .476 | .580 | .647 | .720 | .405 | .474 | **.512** | .622 | .306 | .317 | .296 | .288 | .362 | .340 | .340 | .328 | .438 |

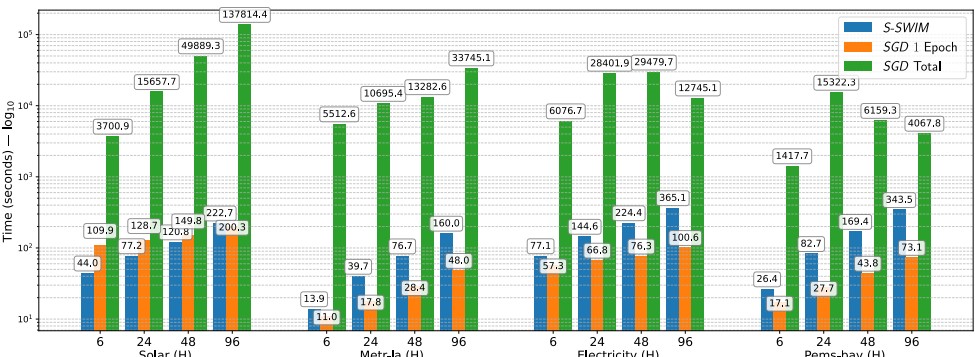

Figure 2: Training time of *S-SWIM* and *SGD* training across datasets and prediction horizons $H$ for the conducted forecasting experiments with the Hat kernel. Note that the time is given in $\log$ scale.

Details on the experimental setup, the employed hardware and datasets are given in section B.

**Time-Series Forecasting:** As the main benchmark, we evaluate the performance of the proposed algorithm for fitting networks on a time series forecasting task. We evaluate the performance of (a) training the networks only with *S-SWIM*, (b) training the networks only with a stochastic gradient descent (*SGD*) algorithm, specifically *ADAM* (Kingma & Ba, 2017), using an established surrogate-gradient strategy, and (c) fine-tuning the *S-SWIM* -trained networks with gradient-based training and compare the results to the current state of the art for *SNN*s. The results can be found in table 1, the corresponding fitting time in figure 2. With the exception of the **Metr-la** dataset, we find that *S-SWIM* consistently performs at or above the level of *SGD*, especially over long prediction horizons. We suspect an interplay between two effects here. Namely, (a) at longer prediction horizons, the credit assignment problem in gradient-descent training naturally gets harder (Pascanu et al., 2013), which *S-SWIM* does not suffer from by being gradient-free and (b) *S-SWIM* seemingly underperforms at very short horizons ($H = 6$ in particular). We suspect that the latter is due to the aggregation of delays in section 3.2.6 (Kernel Supports) not working short horizons. The effect is also

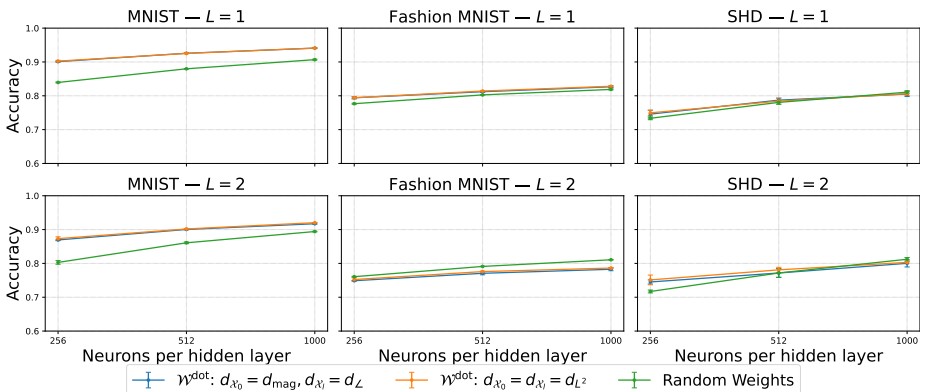

Figure 3: Classification accuracy across different architectures and configurations (Number of hidden layers, number of neurons, weight constructions and sampling metrics (cf. section A.1)).

observed for *SGD*-only training (Hat, **Solar**, $H = 48, 96$), so it must be in part attributable to the model being able to exploit periodic structure in the data (Lai et al. (2018)[Figure 3]) better at long horizons. Finally, the current results suggest that *S-SWIM* can also serve as an efficient but effective initialisation strategy for gradient-based training, however, however, more experiments in this direction are required due to different parameters and non-convergence in multiple cases. Across different algorithms, we also observe that the choice of the employed kernel function(s) matters. Most importantly, *S-SWIM* is extremely fast compared to *SGD* training, consistently achieving speedups of one to three orders of magnitude (cf. figure 2). **Classification Experiments:** We conduct additional experiments on static image and speech classification datasets, including the mutlti-layer case and problems where the targets are not real-valued functions and thus require surrogate target voltages (cf. sections A.4), . We chose those benchmarks as they contain spikes generated from non-spiking data with no real temporal information and "real" spike data with meaningful temporal patterns. The predicted class is taken as the output layer neuron with the largest voltage integrated over time or number of spikes. The accuracy for the voltage readout is shown in figure 3. For the spike-based readout, see section C.1. The findings show that *S-SWIM* can also be used for classification problems and that with the notable exception of $L = 2$ on F-MNIST typically significantly outperforms data-agnostic weights, especially it never performs significantly worse. Regarding sampling deep networks, we find that while those networks still achieve acceptable results, they perform worse than their shallow counterparts, which currently constitutes a major weakness of *S-SWIM* . Further investigation is needed to relate this to poorly chosen hyperparameters, especially the temporal parameters or the sampling metrics or more general problems of the algorithm. It should be noted that this is a common problem of *RFM*.

## 5 CONCLUSION

**Summary:** We introduced *S-SWIM*, a novel data-driven algorithm for sampling the weights of *SNN*s. Our approach addresses key challenges in training *SNN*s by circumventing the approximation of the spike function gradient entirely, through the use of the *RFM* paradigm. Through numerical experiments, we show that *S-SWIM* can serve as a solid standalone strategy or initialisation for gradient-based training, especially at long prediction horizons, while being orders of magnitude faster than full *SGD* optimisation. Furthermore, the approach is interpretable and modular, easing future analysis, refinement and adaptation to different tasks. **Limitations:** *S-SWIM* does currently not perform well for deep networks, barring it from practical applicability. It is currently unclear, why. Several avenues for improvement remain unexplored. See section D for further discussion.

## 6 ETHICS STATEMENT

This work contributes to the advancement of spiking neural networks, which are regarded as a potentially very energy-efficient family of machine learning models. Our main contribution is towards making the training of spiking neural networks faster, thus saving resources, and making it more interpretable. While they are broadly applicable and can thus potentially be misused, we see the benefits of reducing the immense cost of training for proper use as outweighing this risk. Especially since there are growing concerns about the resource demand of artificial neural networks for training, which is addressed by our method, and inference, which is addressed by the broader field of spiking neural networks.

## 7 REPRODUCIBILITY STATEMENT

The authors have worked diligently toward ensuring the reproducibility of the results presented in this work. Specifically, we give a thorough discussion of the algorithm in section 3.2 and section A, enabling independent implementation and evaluation of the proposed method. A comprehensive description of the setup for the main experiments is given in section B. Furthermore, the code used for this study is provided in the supplementary materials and will be made publicly available after the review period.

The datasets used for the experiments can be downloaded through an anonymised link provided as part of the supplementary materials.

## 8 USAGE OF *LLM*S & *AI* TOOLS

Several *LLM / AI* tools have been used throughout various stages of this work, which we will detail here.

**AI-assistance during code development.**    During the development of the code implementing the proposed method for the numerical experiments, *LLM*s were employed to assist the process. Concretely, the models ChatGPT (OpenAI), Claude (Anthropic), and DeepSeek LLM (DeepSeek-AI) were used. The major developments of the code happened between March 2025 and August 2025, during which the most recent publicly available versions of the models were used. All code outputs generated by these models were manually reviewed and, where necessary, corrected or adapted by the authors to ensure correctness and reproducibility.

**AI-assistance during paper writing.**    Throughout all sections of the paper, the *AI* tool Grammarly (Grammarly, Inc.) was used for correcting spelling and grammar. Some sentences were rephrased according to the suggestions of Grammarly. All suggestions made by Grammarly were reviewed and only accepted if deemed correct.

The authors acknowledge the full responsibility for the submitted text, code and presented results, especially for any mistakes contained therein.

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

# A   DETAILED DERIVATIONS

Here, we provide more details about the theoretical underpinnings of the *S-SWIM* algorithm.

## A.1   SAMPLING DISTRIBUTION

**Pseudometrics.**   We propose to use pseudometrics for sampling, that is a non-negative functions of the form $d : X \times X \mapsto \mathbb{R}_0^+$, for some set $X$, fulfilling $\forall x, y, z \in X$

$$d(x, x) = 0, \tag{11}$$

$$d(x, y) = d(y, x), \tag{12}$$

$$d(x, z) \le d(x, y) + d(y, z). \tag{13}$$

We intentionally do not require $x \ne y \implies d(x, y) > 0$, as it can be useful to identify samples differing only in a way that is not informative for sampling. Below we will define the pseudometrics we evaluated during our numerical experiments, which for example identify a function with scalar multiples or phase-shifted copies of itself.

**General Construction.**   We suggest the following general construction for informative sampling pseudometics. Let $\mathcal{H} = L^2\left([a, b], \mathbb{R}^J\right)$, $\mathcal{E} = L^2\left([a, b], \mathbb{C}^J\right)$ be equipped with inner products $\langle x, y \rangle_{\mathcal{H}} = \int_a^b x(t)^\top y(t) \, \mathrm{d}t$, respectively, $\langle x, y \rangle_{\mathcal{E}} = \int_a^b x(t)^H y(t) \, \mathrm{d}t$, where $x(t)^H$ is the conjugate transpose of $x(t)$, and their induced norms $\|\cdot\|_{\mathcal{H}}, \|\cdot\|_{\mathcal{E}}$. Pseudometrics on $\mathcal{H}$ can now be defined by choosing a suitable embedding $E : \mathcal{H} \mapsto \mathcal{E}$ through the construction

$$d_{\mathcal{H}}(x, y) = \|E(x) - E(y)\|_{\mathcal{E}} = \left( \int_a^b \|E(x)(t) - E(y)(t)\|_{\mathbb{C}^J}^2 \, \mathrm{d}t \right)^{1/2}. \tag{14}$$

**Proposition A.1.** $d_{\mathcal{H}}$ *is a pseudo-metric on* $\mathcal{H}$.

*Proof.* 11, and 12 are obvious. 13 follows from the Minkowski inequality, as

$$\|E(x) - E(z)\|_{\mathcal{E}} = \|E(x) - E(y) + E(y) - E(z)\|_{\mathcal{E}} \le \|E(x) - E(y)\|_{\mathcal{E}} + \|E(y) - E(z)\|_{\mathcal{E}}. \tag{15}$$
$$\square$$

This construction generalises the notion of the canonical $L^2$ distance by only measuring distance on properties that are relevant for the purpose of constructing weights, effectively identifying samples differing only in irrelevant characteristics. This construction can naturally be extended to spike-valued inputs or outputs by first embedding the target spike-trains into $\mathcal{H}$. For this, we apply the idea of the *van Rossum* metric on spike trains (van Rossum, 2001). Concretely, define the set of all $D$-dimensional spike trains of arbitrary but finite length over the interval $I$,

$$\mathcal{S}_I^D = \bigtimes_{d=1}^{D} \left\{ \sum_{i=1}^{N} \delta(\cdot - t_i^f) \,\middle|\, t_i^f \in I \wedge N < \infty \right\}, \tag{16}$$

and the induced pseudometric $d_S$ over $\mathcal{S}_I^D$ from a pseudometric $d_f$ over a space of real-valued functions through

$$d_S(S_1, S_2) = d_f\left([S_1 * h], [S_2 * h]\right). \tag{17}$$

The kernel $h$ is in principle arbitrary as long as a pseudometric $d^f$ can be defined. A reasonable choice in the context of *S-SWIM* when sampling for layer $l$ is the *SRK* $k^{(l)}$, which by finity of $S$ and integrability of $k^{(l)}$ yields $[S * k^{(l)}] \in L^2\left(I, \mathbb{R}^D\right)$ for $S \in \mathcal{S}_I^D$, where the convolution is performed per dimension.

To make the sampling invariant to constant shifts, the dimensionwise temporal mean,

$$\mu_{x_i}(a, b) = \frac{1}{b - a} \int_a^b x_i(t)\, \mathrm{d}t, \tag{18}$$

is subtracted from all samples before computing the pairwise distances, which can be absorbed into the embedding in the presented framework. Furthermore, we impose a small dimensionwise minimum $L^2$ norm to each input function, setting the probability of pairs containing samples which do not fulfil this criterion to 0, as such pairs yield trivial and thus not useful solutions for the discussed weight constructions.

Other definitions of "distance" can of course be used instead. We chose this construction mainly because it for one is simple yet flexible, making theoretical analysis easier, and because it is informative in the sense that which embeddings $E$ produce informative gradients yields insights into the structure of the data.

**Example Embeddings.** Several useful embeddings that together with equation 17 can be used to construct the pseudometrics $d_{\tilde{\mathcal{X}}_0}, d_{\tilde{\mathcal{X}}_l}, d_{\tilde{\mathcal{Y}}}$ by appropriate choices of $a, b$ and $J$ in the above definitions and denoting the fourier transform of $f$ as $\mathcal{F}\{f\}$ are

- $E_{L^2} = \mathrm{Id}$, which yields the standard $L^2$ metric,

- $E_{\cos}(x) = \begin{cases} \dfrac{x}{\|x\|_{L^2}}, & x \neq 0, \\ 0, & x = 0, \end{cases}$ which identifies real scalar multiples and thus is only sensitive to global shape (dis-)similarity,

- $E_{\mathrm{mag}}(x, \omega) = \left| \mathcal{F}\{x\}(\omega) \right|$, which is insensitive to phase shifts,

- $E_{\angle}(x, \omega) = \begin{cases} \dfrac{\mathcal{F}\{x\}(\omega)}{|\mathcal{F}\{x\}(\omega)|}, & \mathcal{F}\{x\}(\omega) \neq 0, \\ 0, & \mathcal{F}\{x\}(\omega) = 0, \end{cases}$ which only measures relative phase structure, and

- $E_B(x, \omega) = \mathbf{1}_K(\omega)\mathcal{F}\{x\}(\omega)$, where $\mathbf{1}_B$ is the indicator function, if only differences in a specific band of frequencies (say only high for fast fluctuations or only low for slow trends) are considered important.

**Entropy Criterion.** The pseudometrics can either by specified manually using prior knowledge or assumptions about the data, or chosen automatically based on a suitable heuristic. With $\mathcal{D}_{\mathrm{in}}, \mathcal{D}_{\mathrm{out}}$ being sets of candidate pseudometrics for the inputs and outputs respectively, we propose to choose a small subset $X_s$ of training samples, and select

$$d_{\mathrm{in}}, d_{\mathrm{out}} = \operatorname*{arg\,min}_{d_{\mathrm{in}} \in \mathcal{D}_{\mathrm{in}}, d_{\mathrm{out}} \in \mathcal{D}_{\mathrm{out}}} \mathbf{H}(P_{d_{\mathrm{in}}, d_{\mathrm{out}}}) \tag{19}$$

over $X_s$, where $P_{d_{\mathrm{in}}, d_{\mathrm{out}}}$ is the normalised probability distribution used for sampling in algorithm 1 and computed with respect to the pseudometrics $d_{\mathrm{in}}$ and $d_{\mathrm{out}}$. $\mathbf{H}(p)$ denotes the Shannon entropy (Shannon, 1948) defined as

$$\mathbf{H}(p) = -\sum_{a \in A} p(a) \log p(a) \tag{20}$$

for a probability distribution $p$ over the elements of $A$. The base of the logarithm is arbitrary in our case. The entropy is maximised by the uniform distribution, so it can be used to measure the non-uniformity of a distribution. As explained below, the weights of the hidden layer(s) are chosen to maximise some notion of dissimilarity between the latent embeddings of sample pairs with large gradients. Thus, we want to select exactly those feature embeddings (among sensibly chosen candidates) which yield large gradients, or in

other words, a natural way to separate the given dataset[1]. The entropy criterion has several particularly nice properties:

1. It has analytically available bounds from below $(0)$ and above $(\log|A|)$.

2. It is an intrinsic property of the data, i.e. independent of any chosen network architecture.

3. It is invariant to different ranges of values assigned by different pseudometrics, since it uses normalised probabilities.

4. It is relatively cheap to compute even for a moderate number of candidates. Specifically, only $|\mathcal{D}_{\text{in}}| + |\mathcal{D}_{\text{out}}|$ pairwise distance computations are required, as the distance matrices can be reused.

## A.2 SAMPLED WEIGHTS

**Derivation of the proposed weights.** Letting $x_{i,1}^{(l)}, x_{i,2}^{(l)} \in \mathbb{R}^{N_{l-1}}$ be the inputs sampled from $P^{(l)}$ for neuron $i$ and defining the vectors $\psi_{i,1}^{(l)}(t), \psi_{i,2}^{(l)}(t) \in \mathbb{R}^{N_{l-1}}$ elementwise as $\left(\psi_{i,n}^{(l)}(t)\right)_j = \psi_{ij}^{(l)}\left(x_{i,n}^{(l)}, t\right)$, we can write the contribution to the voltage from the inputs $v_{i,n}^{(l)}$ using the euclidean inner product $\langle\cdot,\cdot\rangle_{\ell^2}$ as $v_{i,n}^{(l)}(t) = \left\langle W_{i,:}^{(l)}, \psi_{i,n}^{(l)}(t)\right\rangle_{\ell^2}$ for $n \in \{1,2\}$. Since $v_{i,n}^{(l)} \in L^2([a,b),\mathbb{R})$, as by the assumptions $\psi_{ij}^{(l)}\left(x_{i,n}^{(l)}\right) \in L^2([a,b),\mathbb{R})$, we can use the $L^2$ inner product $\langle\cdot,\cdot\rangle_{L^2}$ and its induced metric $d_{L^2}(\cdot,\cdot)$ to formalise the requirement of separation in function space. The proposed objectives can then be stated as

$$\max_{\|w\|=1} d_{L^2}\left(v_{i,1}^{(l)}, v_{i,2}^{(l)}\right) = \max_{\|w\|=1} \int_a^b \left(\left\langle w, \psi_{i,1}^{(l)}(t)\right\rangle_{\ell^2} - \left\langle w, \psi_{i,2}^{(l)}(t)\right\rangle_{\ell^2}\right)^2 \mathrm{d}t \tag{21}$$

and

$$\min_{\|w\|=1} \left\langle v_{i,1}^{(l)}, v_{i,2}^{(l)}\right\rangle_{L^2} = \min_{\|w\|=1} \int_a^b \left\langle w, \psi_{i,1}^{(l)}(t)\right\rangle_{\ell^2} \cdot \left\langle w, \psi_{i,2}^{(l)}(t)\right\rangle_{\ell^2} \mathrm{d}t. \tag{22}$$

The bounds are respectively $a = 0$ and $b = T + H$ in algorithm 1.

**Proposition A.2.** *The solution to the optimisation problem 21 (22) is given by the eigenvector $w_{\max}$ ($w_{\min}$) to the algebraically largest (smallest) eigenvalue $\lambda_{\max}$ ($\lambda_{\min}$) of a symmetric matrix $A \in \mathbb{R}^{N_{l-1} \times N_{l-1}}$. Further, for 21*

$$A^{dist} = \int_a^b \left(\psi_{i,1}^{(l)}(t) - \psi_{i,2}^{(l)}(t)\right)\left(\psi_{i,1}^{(l)}(t) - \psi_{i,2}^{(l)}(t)\right)^\top \mathrm{d}t \in \mathbb{R}^{N_{l-1}}, \tag{23}$$

*and for 22*

$$A^{dot} = \frac{1}{2} \int_a^b \psi_{i,1}^{(l)}(t)\psi_{i,2}^{(l)}(t)^\top + \psi_{i,2}^{(l)}(t)\psi_{i,1}^{(l)}(t)^\top \mathrm{d}t. \tag{24}$$

*Proof.* We will start by showing that **(I.)** the solutions to

$$\max_{\|w\|=1} w^\top Q w \qquad \text{and} \qquad \min_{\|w\|=1} w^\top Q w \tag{25}$$

for $Q \in \mathbb{R}^{n \times n}$ are respectively $w_{\max}$ and $w_{\min}$ of $\text{sym}(Q) := \frac{1}{2}\left(Q + Q^\top\right)$ and then **(II.)** show that 21 and 22 can be written in the form of equation 25, where $\text{sym}(Q)$ will yield the proposed matrices.

---

[1]Of course, degenerate cases, such as the distribution approaching a dirac mass on a single sample pair, are not desirable. This can easily be safeguarded against by selecting the minimising pair above a given minimum entropy, although we found this to not be an issue in our numerical experiments.

**(I.)** *Extrema of quadratic form.*

*(a)* Let $\lambda$ be a lagrange multiplier. The Lagrangian for equation 25 is then

$$\mathcal{L}(w,\lambda) = w^\top Q w + \lambda\left(w^\top w - 1\right). \tag{26}$$

$$\frac{\partial}{\partial w}\mathcal{L}(w,\lambda) = \left(Q + Q^\top\right)w + 2\lambda w \overset{!}{=} 0 \tag{27}$$

$$\iff \frac{1}{2}\left(Q + Q^\top\right)w = \mathrm{sym}(Q)w = \lambda w, \tag{28}$$

which we recognise as the eigenvalue equation, thus together with the norm constraint making the candidate solutions the eigenvectors with unit norm $\{w_i\}_{i=1}^n$ of $\mathrm{sym}(Q)$.

*(b)* Next, noting that

$$w_i^\top Q w_i = w_i^\top\left(\frac{1}{2}\left(Q + Q^\top\right) + \frac{1}{2}\left(Q - Q^\top\right)\right)w_i \tag{29}$$

$$= w_i^\top \mathrm{sym}(Q)w_i + \frac{1}{2}w_i^\top Q w_i - \frac{1}{2}w_i^\top Q^\top w_i \tag{30}$$

$$= w_i^\top \mathrm{sym}(Q)w_i = \lambda_i w_i^\top w_i = \lambda_i, \tag{31}$$

we get that the solutions to equation 25 are respectively $w_{\max}$ and $w_{\min}$.

**(II.)** *21 and 22 as quadratic forms.*

*(a)* For 21, we have

$$\max_{\|w\|=1} d_{L^2}\left(v_{i,1}^{(l)}, v_{i,2}^{(l)}\right) = \max_{\|w\|=1}\int_a^b\left(\left\langle w, \psi_{i,1}^{(l)}(t)\right\rangle_{\ell^2} - \left\langle w, \psi_{i,2}^{(l)}(t)\right\rangle_{\ell^2}\right)^2 \mathrm{d}t \tag{32}$$

$$= \max_{\|w\|=1}\int_a^b\left(\left\langle w, \psi_{i,1}^{(l)}(t) - \psi_{i,2}^{(l)}(t)\right\rangle_{\ell^2}\right)^2 \mathrm{d}t \tag{33}$$

$$= \max_{\|w\|=1}\int_a^b\left(w^\top\left(\psi_{i,1}^{(l)}(t) - \psi_{i,2}^{(l)}(t)\right)\right)^2 \mathrm{d}t \tag{34}$$

$$= \max_{\|w\|=1}\int_a^b w^\top\left(\psi_{i,1}^{(l)}(t) - \psi_{i,2}^{(l)}(t)\right) w^\top\left(\psi_{i,1}^{(l)}(t) - \psi_{i,2}^{(l)}(t)\right) \mathrm{d}t \tag{35}$$

$$= \max_{\|w\|=1} w^\top \underbrace{\int_a^b\left(\psi_{i,1}^{(l)}(t) - \psi_{i,2}^{(l)}(t)\right)\left(\psi_{i,1}^{(l)}(t) - \psi_{i,2}^{(l)}(t)\right)^\top \mathrm{d}t}_{Q=\mathrm{symm}(Q)=A^{\mathrm{dist}}} w. \tag{36}$$

*(b)* Similarly, for 22, we have

$$\min_{\|w\|=1}\left\langle v_{i,1}^{(l)}, v_{i,2}^{(l)}\right\rangle_{L^2} = \min_{\|w\|=1}\int_a^b\left\langle w, \psi_{i,1}^{(l)}(t)\right\rangle_{\ell^2}\cdot\left\langle w, \psi_{i,2}^{(l)}(t)\right\rangle_{\ell^2} \mathrm{d}t \tag{37}$$

$$= \min_{\|w\|=1}\int_a^b w^\top\psi_{i,1}^{(l)}(t)w^\top\psi_{i,2}^{(l)}(t)\,\mathrm{d}t \tag{38}$$

$$= \min_{\|w\|=1} w^\top\underbrace{\int_a^b\psi_{i,1}^{(l)}(t)\psi_{i,2}^{(l)}(t)^\top \mathrm{d}t}_{Q} w. \tag{39}$$

Now taking the symmetric part yields the proposed matrix

$$\text{symm}(Q) = A^{\text{dot}} = \frac{1}{2} \int_a^b \psi_{i,1}^{(l)}(t)\psi_{i,2}^{(l)}(t)^\top + \psi_{i,2}^{(l)}(t)\psi_{i,1}^{(l)}(t)^\top \, dt. \tag{40}$$

Together with **(I.)**, this completes the proof.

$\square$

**Corollary A.3.** *As metrics are non-negative and by **(I.)** the eigenvalues are the evaluation of the metric at the candidate solutions, all eigenvalues of $A^{dist}$ must be non-negative, making the algebraically largest eigenvalue also the largest in magnitude.*

By corollary A.3, 21 is easy to solve. While finding the algebraically smallest eigenvalue as required for solving 22 is not as easy, it can also be solved very efficiently by specialised algorithms such as *PRIMME* (Stathopoulos & McCombs, 2010). During our numerical experiments, we found that computing all eigenvectors through the optimised `cupy.linalg.eigh` *CuPy* routine directly on the GPU and discarding was equally as fast as transferring the data to the CPU and running *PRIMME*, likely since the individual eigenproblems are rather small for moderate input dimensions and neuron counts.

**Intuition for the resulting weight vectors.** Finally, we will give an intuitive explanation for how the weights resulting from the two criteria relate to each other and the input signals. Looking closely at the general structure of $A^{\text{dist}}$ and $A^{\text{dot}}$, we find that they contain the pairwise $L^2$ inner products of the dimensions of the functions they are constructed from. Letting $f, g$ be some vector-valued functions, we get

$$\int_a^b f(t)g(t)^\top \, dt = \begin{bmatrix} \int_a^b f_1(t)g_1(t)\,dt & \int_a^b f_1(t)g_2(t)\,dt & \cdots & \int_a^b f_1(t)g_n(t)\,dt \\ \int_a^b f_2(t)g_1(t)\,dt & \int_a^b f_2(t)g_2(t)\,dt & \cdots & \int_a^b f_2(t)g_n(t)\,dt \\ \vdots & \vdots & \ddots & \vdots \\ \int_a^b f_n(t)g_1(t)\,dt & \int_a^b f_n(t)g_2(t)\,dt & \cdots & \int_a^b f_n(t)g_n(t)\,dt \end{bmatrix} \tag{41}$$

$$= \begin{bmatrix} \langle f_1, g_1 \rangle_{L^2} & \langle f_1, g_2 \rangle_{L^2} & \cdots & \langle f_1, g_n \rangle_{L^2} \\ \langle f_2, g_1 \rangle_{L^2} & \langle f_2, g_2 \rangle_{L^2} & \cdots & \langle f_2, g_n \rangle_{L^2} \\ \vdots & \vdots & \ddots & \vdots \\ \langle f_n, g_1 \rangle_{L^2} & \langle f_n, g_2 \rangle_{L^2} & \cdots & \langle f_n, g_n \rangle_{L^2} \end{bmatrix}. \tag{42}$$

Thus, we see that $\mathcal{W}^{\text{dist}}$ gives the vector which is most aligned with the dimensions where the convolved input signals differ, whereas $\mathcal{W}^{\text{dot}}$ give the vector least aligned with where the convolved input signals are similar. Furthermore, restating equation 21 as

$$\max_{\|w\|=1} \int_a^b \left( \left\langle w, \psi_{i,1}^{(l)}(t) \right\rangle_{\ell^2} - \left\langle w, \psi_{i,2}^{(l)}(t) \right\rangle_{\ell^2} \right)^2 \, dt \tag{43}$$

$$= \max_{\|w\|=1} \int_a^b \left\langle w, \psi_{i,1}^{(l)}(t) \right\rangle_{\ell^2}^2 \, dt + \int_a^b \left\langle w, \psi_{i,2}^{(l)}(t) \right\rangle_{\ell^2}^2 \, dt - 2 \int_a^b \left\langle w, \psi_{i,1}^{(l)}(t) \right\rangle_{\ell^2} \left\langle w, \psi_{i,2}^{(l)}(t) \right\rangle_{\ell^2} \, dt \tag{44}$$

$$= \max_{\|w\|=1} w^\top \left( \underbrace{\int_a^b \psi_{i,1}^{(l)}(t)\psi_{i,1}^{(l)}(t)^\top \, dt + \int_a^b \psi_{i,2}^{(l)}(t)\psi_{i,2}^{(l)}(t)^\top \, dt}_{\text{magnitudes}} - 2 \underbrace{\int_a^b \psi_{i,1}^{(l)}(t)\psi_{i,2}^{(l)}(t)^\top \, dt}_{\text{dot}} \right) w, \tag{45}$$

we find that $\mathcal{W}^{\text{dist}}$ is a compromise between $\mathcal{W}^{\text{dot}}$ and aligning with input dimensions having high amplitudes, in other words, aligning with the dimensions where the signals are large and dissimilar.

This clear interpretability is a major advantage of *S-SWIM* over gradient-based methods.

## A.3 NORMALISATION

**Satisfying the constraints.** Formally, defining the input contribution to a neuron $i$ in layer $l$ from a sample $x^{(n)}$ using the scaled candidate weights $\tilde{w}$ from the sampling step scaled by a factor $\alpha \in \mathbb{R}^+$ and a bias $\beta \in \mathbb{R}$, which remain to be specified, as

$$v_{i,n}^{(l)}(t) = \left\langle \alpha\tilde{w}, \psi_{i,n}^{(l)}(t) \right\rangle_{\ell^2} + \beta, \tag{46}$$

where $\left( \psi_{i,n}^{(l)}(t) \right)_j = \psi_{ij}^{(l)}\left(x^{(n)}, t\right)$, the statistics of interest are the expected temporal mean and standard deviation

$$\mathbb{E}_{v_i}^X = \mathbb{E}_n^X\left[\mathbb{E}_t^X\left[v_{i,n}^{(l)}(t)\right]\right] \qquad \text{and} \qquad \mathbb{S}_{v_i}^X = \mathbb{E}_n^X\left[\sqrt{\mathrm{Var}_t^X\left[v_{i,n}^{(l)}(t)\right]}\right] \tag{47}$$

over the dataset $X$, which we approximate by $\mathbb{E}_{v_i}^{X_I} \approx \mathbb{E}_{v_i}^X, \mathbb{S}_{v_i}^{X_I} \approx \mathbb{S}_{v_i}^X$ on the subset $X_I$. By applying the properties of expectation and variance, we get the two equations

$$\mathbb{E}_{v_i}^{X_I} = \alpha\mathbb{E}_n^{X_I}\left[\mathbb{E}_t^{X_I}\left[\left\langle \tilde{w}, \psi_{i,n}^{(l)}(t) \right\rangle_{\ell^2}\right]\right] + \beta \tag{48}$$

$$\mathbb{S}_{v_i}^{X_I} = \alpha\mathbb{E}_n^{X_I}\left[\sqrt{\mathrm{Var}_t^{X_I}\left[\left\langle \tilde{w}, \psi_{i,n}^{(l)}(t) \right\rangle_{\ell^2}\right]}\right] \tag{49}$$

in the four unknowns $\alpha, \beta, \mathbb{E}_{v_i}^{X_I}$, and $\mathbb{S}_{v_i}^{X_I}$, leaving two degrees of freedom to impose the desired constraints. For explicitly prescribing the target mean $\mathbb{E}_{v_i}^{X_I} \overset{!}{=} \mu_t$ and standard $\mathbb{S}_{v_i}^{X_I} \overset{!}{=} s_t$ (*MS*), we get

$$\alpha_{s_t}^{\mathrm{MS}} = \frac{s_t}{\mathbb{E}_n^{X_I}\left[\sqrt{\mathrm{Var}_t^{X_I}\left[\left\langle \tilde{w}, \psi_{i,n}^{(l)}(t) \right\rangle_{\ell^2}\right]}\right]} \quad \text{and} \quad \beta_{\mu_t,s_t}^{\mathrm{MS}} = \mu_t - s_t\frac{\mathbb{E}_n^{X_I}\left[\mathbb{E}_t^{X_I}\left[\left\langle \tilde{w}, \psi_{i,n}^{(l)}(t) \right\rangle_{\ell^2}\right]\right]}{\mathbb{E}_n^{X_I}\left[\sqrt{\mathrm{Var}_t^{X_I}\left[\left\langle \tilde{w}, \psi_{i,n}^{(l)}(t) \right\rangle_{\ell^2}\right]}\right]} \tag{50}$$

for the remaning unknowns, where $\mu_t \in (-\infty, 1)$ and $s_t \in \mathbb{R}^+$ are the tunable hyperparameters. For the *FL* criterion, we fix the standard deviation

$$\mathbb{S}_{v_i}^{X_I} \overset{!}{=} \mathbb{E}_n^{X_I}\left[\sqrt{\mathrm{Var}_t^{X_I}\left[\left\langle \tilde{w}, \psi_{i,n}^{(l)}(t) \right\rangle_{\ell^2}\right]}\right] \tag{51}$$

and need to guarantee $\mathbb{E}_{v_i}^{X_I} \overset{!}{=} 1 - r\mathbb{S}_{v_i}^{X_I}$ with $r \in \mathbb{R}^+$ as tunable parameter. This is satisfied by assigning $\alpha^{\mathrm{FL}} = 1$ and

$$\beta_z^{\mathrm{FL}} = 1 - z\mathbb{E}_n^{X_I}\left[\sqrt{\mathrm{Var}_t^{X_I}\left[\left\langle \tilde{w}, \psi_{i,n}^{(l)}(t) \right\rangle_{\ell^2}\right]}\right] - \mathbb{E}_n^{X_I}\left[\mathbb{E}_t^{X_I}\left[\left\langle \tilde{w}, \psi_{i,n}^{(l)}(t) \right\rangle_{\ell^2}\right]\right] \tag{52}$$

to the remaining quantities. The *MS* criterion gives more explicit control over how much the spiking frequency is determined by input fluctuations and how much by a regular baseline, whereas the *FL* criterion is fully driven by fluctuations but only has one parameter that potentially needs tuning. In both cases, the observed average voltage will of course be lower than the prescribed voltage since the above computations neglect the spike cost. These deviations will however be systematic and somewhat limited by the self-regulating nature of the spiking mechanism, making the used statistics nonetheless meaningful approximations to the actually observed voltage.

**Silence Correction.** Furthermore, to guarantee that every neuron actually spikes at least once (on $X_I$), $\beta$ can be post-corrected by additionally tracking the per-neuron maximum voltage

$$\mathbb{M}_{v_i}^{X_I} = \max_t {}^{X_I} \left[ \max_n {}^{X_I} \left[ \left\langle \tilde{w}, \psi_{i,n}^{(l)}(t) \right\rangle_{\ell^2} \right] \right]. \tag{53}$$

The term correcting the bias $\Delta^{\text{SC}}$ to avoid silent ("*SC*") neurons is then given by

$$\Delta^{\text{SC}} = \begin{cases} (1 + \varepsilon) - \mathbb{M}_{v_i}^{X_I} & \text{for } \mathbb{M}_{v_i}^{X_I} < 1, \\ 0 & \text{otherwise,} \end{cases} \tag{54}$$

where the small constant $\varepsilon \geq 0$ is added for numerical robustness. This ensures that, when using the updated bias $\beta + \Delta^{\text{SC}}$, $\tilde{\mathbb{M}}_{v_i}^{X_I} \geq 1$, meaning at least one spike will be emitted. A similar construction can be derived for $\alpha$.

**Computing the statistics.** It should be noted that all three required statistics can jointly be computed in a single pass over $X_I$, even if $X_I$ does not fit into the available memory, by online algorithms such as Welford's algorithm (Welford, 1962), and using that the new maximum after normalisation will be $\tilde{\mathbb{M}}_{v_i}^{X_I} = \alpha \mathbb{M}_{v_i}^{X_I} + \beta$ (since $\alpha > 0$), as thus computing the pre-normalisation maximum is sufficient.

### A.4 SPIKE-VALUED TARGETS

The initialisation of the hidden layers works essentially the same for real- and spike-valued targets. Throughout the discussion of the parameters in the output layer, we assume that $Y$ is comprised of real-valued functions, which we prescribe as the target voltages for that layer. In the case of spike-valued outputs, surrogate voltages need to be generated for each neuron $i$ that could produce the target spike trains and lie in the linear span of $\{\psi_{ij}^{(l)}\}_{j=1}^{N_L} \cup \{1, \eta_i^{(l)}\}$ respectively applied to the associated inputs and target trains. The constraints for exactly producing a given spike-train are given by $v^{\text{surr}}(t^f) = 1$ for all firing times $f^f$ in the target train and $v^{\text{surr}}(t) < 1$ otherwise. It is likely that these constraints can't be jointly fulfilled for all samples and thus would need to be relaxed suitably.

For example, for classification a natural surrogate is to set the targets to a positive constant above the firing threshold for the ground-truth class and to a negative constant for the other classes.

While this idea is straightforward, it is far from trivial to apply to more complex cases, which is why we defer a thorough investigation to future work.

Alternatively, it has been shown that optimising the linear parameters $\boldsymbol{W}, \boldsymbol{b}, \boldsymbol{c}$ of a single *SRM*-layer for spike-valued outputs can be solved by Poisson Generalised Linear Model regression (Truccolo et al., 2005; Weber & Pillow, 2017). This, however, requires costly iterative optimisation and it is unclear how to fit the temporal parameters $\boldsymbol{\tau}, \boldsymbol{\sigma}, \boldsymbol{\varsigma}$, which contribute non-linearly to the output, in this case.

### A.5 OUTPUT DELAYS

The purpose of the delays is to select the interval from the output of the hidden layers best suited for constructing the target function, more specifically the time-varying part, from the *SRK*s placed at the spikes inside this interval. Let $y_i^{(n)}$ denote the target output of sample $n$ for neuron $i$, $\tilde{y}_i^{(n)} = y_i^{(n)} - \mu_{y_i^{(n)}}(T, T + H)$ its time-varying part, and $S_j^{(n)} = \Psi_j^{(L)}\left(x^{(n)}\right)$ the output spike train of neuron $j$ in the last hidden layer to the associated input sample.

Assuming the ansatz of definition 3.1 for $\tilde{y}_i^{(n)}$ and using the shorthand

$$k_i^{(L+1)}(t) = k^{(L+1)} \left( \frac{t - \tau_i^{(L+1)}}{\sigma_i^{(L+1)}} \right) \Bigg|_{t \geq 0}, \tag{55}$$

we set

$$y_i^{(n)} \overset{!}{=} \sum_{j=1}^{N_L} W_{ij}^{(L+1)} \left[ k_i^{(L+1)} * S_j^{(n)} \right] + b_i^{(L+1)} \tag{56}$$

yielding

$$\tilde{y}_i^{(n)} = y_i^{(n)} - \frac{1}{H} \int_T^{T+H} \sum_{j=1}^{N_L} W_{ij}^{(L+1)} \left[ k_i^{(L+1)} * S_j^{(n)} \right] (t) + b_i^{(L+1)} \, \mathrm{d}t \tag{57}$$

$$= \sum_{j=1}^{N_L} W_{ij}^{(L+1)} \left[ k_i^{(L+1)} * S_j^{(n)} \right] - \int_T^{T+H} \left[ k_i^{(L+1)} * S_j^{(n)} \right] (t) \, \mathrm{d}t \tag{58}$$

$$= \sum_{j=1}^{N_L} W_{ij}^{(L+1)} \left[ \left( k_i^{(L+1)} - \mu_{k_i^{(L+1)}}(T, T+H) \right) * S_j^{(n)} \right] \tag{59}$$

$$:= \sum_{j=1}^{N_L} W_{ij}^{(L+1)} \left[ \hat{k}_i^{(L+1)} * S_j^{(n)} \right]. \tag{60}$$

While the parameters $W_i^{(L+1)}$ and $\sigma_i^{(L+1)}$ here are unknown, we can get an independent estimate for the delay using correlation analysis. We find that

$$C_{ij}^{(n)}(\tau) = \left[ S_j^{(n)} \star \tilde{y}_i^{(n)} \right] = \sum_{t_j^{(n)} \in T_j^{(n)}} \tilde{y}_i^{(n)} \left( \tau - t_j^{(n)} \right) \tag{61}$$

$$= \sum_{m=1}^{N_L} W_{im}^{(L+1)} \sum_{t_j^{(n)} \in T_j^{(n)}} \left[ \hat{k}_i^{(L+1)} * S_m^{(n)} \right] \left( \tau - t_j^{(n)} \right) \tag{62}$$

$$= \sum_{m=1}^{N_L} W_{im}^{(L+1)} \sum_{t_j^{(n)} \in T_j^{(n)}} \sum_{t_m^{(n)} \in T_m^{(n)}} \hat{k}_i^{(L+1)} \left( \tau - t_j^{(n)} - t_m^{(n)} \right), \tag{63}$$

which after separating the $m = j \wedge t_j^{(n)} = t_m^{(n)}$ and $m \neq j \vee t_j^{(n)} \neq t_m^{(n)}$ terms gives

$$C_{ij}^{(n)}(\tau) = \left| T_j^{(n)} \right| W_{ij}^{(L+1)} \hat{k}_i^{(L+1)}(\tau) \tag{64}$$

$$+ \underbrace{\sum_{m=1}^{N_L} W_{im}^{(L+1)} \sum_{t_j^{(n)} \in T_j^{(n)}} \sum_{\substack{t_m^{(n)} \in T_m^{(n)} \\ m \neq j \vee t_j^{(n)} \neq t_m^{(n)}}} \hat{k}_i^{(L+1)} \left( \tau - t_j^{(n)} - t_m^{(n)} \right)}_{\approx H \mu_{\hat{k}_i^{(L+1)}}(T, T+H) = 0}$$

$$\approx \left| T_j^{(n)} \right| W_{ij}^{(L+1)} \hat{k}_i^{(L+1)}(\tau). \tag{65}$$

The final approximation assumes that the differences between spikes across trains are distributed roughly uniformly over the interval, which is justified if the spike trains of different neurons are only weakly correlated, which is likely since each neuron in the hidden layers has different temporal parameters. The remaining, or at least dominant, term reveals the unknown delay $\tau_i^{(L+1)}$. Specifically, if $k^{(L+1)}$ peaks at $\Delta^k$, $\hat{k}_i^{(L+1)}(\tau)$ will peak at $\tau_i^{(L+1)} + \Delta^k$, contributing a distinct positive or negative extremum to $C_{ij}^{(n)}$, depending on the sign of $W_{ij}^{(L+1)}$. Thus, if we aggregate $\left| C_{ij}^{(n)} \right|$ across many samples and trains, we accumulate exactly those peaks, meaning

$$\tau_i^{(L+1)} = \arg\max_{\tau \in [0,O)} C_i^{(\text{abs})}(\tau) - \Delta^k = \arg\max_{\tau \in [0,O)} \sum_n \sum_{j=1}^{N_L} \left| C_{ij}^{(n)}(\tau) \right| - \Delta^k, \tag{66}$$

as long as the above approximation holds for most trains in most samples. The bound is chosen to guarantee a sufficiently large interval of inputs as previously discussed in section 3.2.1. The correlations can be computed efficiently using equation 61, again utilising spike sparsity, or by employing the Cross-Correlation Theorem, since the transforms $\mathcal{F}\left\{ S_j^{(n)} \right\}$ can be reused across neurons in the output layer. The aggregation over samples $n$ in equation 66 could in principle be done over the entire training data, however we found aggregating only over $X_I$ to already yield a robust estimator.

### A.6 OUTPUT KERNEL SUPPORTS

**Delay Aggregation.** Based on observations from numerical experiments, we make the assumptions that, (a) for moderate values of $H$, the variation between the delays of most neurons is generally smaller than $H$, which is further discussed in section 4 (Numerical Experiments), (b) the larger $\sigma$, the more similar similar values of $\sigma$ perform, i.e. small differences only matter for small values, and (c) $\sigma$ is stationary over time and samples, meaning that if enough samples over a large enough interval are considered, most subbatches will yield similar values. Using this, we pick a representative aggregate of the delay, such as

$$\bar{\tau} = \operatorname*{median}_i \tau_i^{(L+1)} \qquad \text{or} \qquad \bar{\tau} = \min_i \tau_i^{(L+1)} \tag{67}$$

and a set of appropriately spaced candidates $\{\sigma_m^c\}_{m=1}^{N_\sigma^o}$ and simply evaluate how well they perform on the subsets $X_I$ and $Y_I$. The crucial insights making the direct search approach feasible, even cheap, are that aggregating the delays leads to the augmented design matrix in the linear problem being shared by all neurons in the output layer and that, especially after discretisation, only a small set of values needs to be searched. Furthermore, each candidate can be evaluated efficiently, since we only need to compute the residual norm with respect to the best weights for a given $\sigma^c$ but not the actual weights, which we will compute on the full dataset in the final substep.

**Optimal residual.** Concretely, defining $\mathcal{D}\psi_{j|\sigma^c}^{(n)} \in \mathbb{R}^{H_D}$ to be an arbitrary discretisation of the *SRK* contributions from input $j$, evaluated using $\bar{\tau}$ and $\sigma^c$, and $\mathcal{D}y_i^{(n)} \in \mathbb{R}^{H_D}$ a respective discretisation of $y_i^{(n)}$, the augmented design matrix $\boldsymbol{A}(\sigma^c) \in \mathbb{R}^{(H_D \cdot \tilde{M}) \times (N_L + 1)}$ is given by

$$\boldsymbol{A}(\sigma^c) = \begin{bmatrix} \mathbf{1}_{H_D} & \mathcal{D}\psi_{1|\sigma^c}^{(1)} & \mathcal{D}\psi_{2|\sigma^c}^{(1)} & \cdots & \mathcal{D}\psi_{N_L|\sigma^c}^{(1)} \\ \mathbf{1}_{H_D} & \mathcal{D}\psi_{1|\sigma^c}^{(2)} & \mathcal{D}\psi_{2|\sigma^c}^{(2)} & \cdots & \mathcal{D}\psi_{N_L|\sigma^c}^{(2)} \\ \vdots & \vdots & \vdots & \ddots & \vdots \\ \mathbf{1}_{H_D} & \mathcal{D}\psi_{1|\sigma^c}^{(\tilde{M})} & \mathcal{D}\psi_{2|\sigma^c}^{(\tilde{M})} & \cdots & \mathcal{D}\psi_{N_L|\sigma^c}^{(\tilde{M})} \end{bmatrix}, \tag{68}$$

where $\mathbf{1}_{H_D} \in \mathbb{R}^{H_D}$ is a vector of ones added to incorporate the bias term. Similarly concatenating $\mathcal{D}\boldsymbol{y}_i^{(n)}$ over samples $n$ yields the associated target vector $\mathcal{D}\boldsymbol{y}_i$ giving rise to the overdetermined linear problem

$$\boldsymbol{p}_{i|\sigma_c}^* = \arg\min_{\boldsymbol{p}} \left\| \boldsymbol{r}_{i|\sigma_c} \right\|_2^2 = \arg\min_{\boldsymbol{p}} \left\| \boldsymbol{A}(\sigma^c)\boldsymbol{p} - \mathcal{D}\boldsymbol{y}_i \right\|_2^2. \tag{69}$$

We will discuss the theoretical implications of treating time points like samples in section A.7 (Output Weights) when detailing how the final weights are solved for. Using the normal equations, the least squares optimal parameters of neuron $i$ are given in closed form by

$$\boldsymbol{p}_{i|\sigma_c}^* = \begin{pmatrix} b_{i|\sigma_c}^* & \boldsymbol{w}_{i|\sigma_c}^* \end{pmatrix} = \left( \boldsymbol{A}(\sigma^c)^\top \boldsymbol{A}(\sigma^c) \right)^{-1} \boldsymbol{A}(\sigma^c)^\top \mathcal{D}\boldsymbol{y}_i. \tag{70}$$

This expression can be simplified by substituting $\boldsymbol{A}(\sigma^c)$ by its thin $QR$-factorisation $\boldsymbol{A}(\sigma^c) = \boldsymbol{Q}_{\sigma^c}\boldsymbol{R}_{\sigma^c}$, where $\boldsymbol{Q}_{\sigma^c} \in \mathbb{R}^{(H_D \cdot \bar{M}) \times m}$, $m = \operatorname{rank}\left( \boldsymbol{A}(\sigma^c) \right)$, $\boldsymbol{Q}_{\sigma^c}^\top \boldsymbol{Q}_{\sigma^c} = \mathbf{Id}_m$, and $\boldsymbol{R}_{\sigma^c} \in \mathbb{R}^{m \times m}$ is invertible. Using these properties, equation 70 becomes

$$\boldsymbol{p}_{i|\sigma_c}^* = \left( \boldsymbol{R}_{\sigma^c}^\top \boldsymbol{Q}_{\sigma^c}^\top \boldsymbol{Q}_{\sigma^c} \boldsymbol{R}_{\sigma^c} \right)^{-1} \boldsymbol{R}_{\sigma^c}^\top \boldsymbol{Q}_{\sigma^c}^\top \mathcal{D}\boldsymbol{y}_i = \boldsymbol{R}_{\sigma^c}^{-1} \boldsymbol{R}_{\sigma^c}^{-\top} \boldsymbol{R}_{\sigma^c}^\top \boldsymbol{Q}_{\sigma^c}^\top \mathcal{D}\boldsymbol{y}_i = \boldsymbol{R}_{\sigma^c}^{-1} \boldsymbol{Q}_{\sigma^c}^\top \mathcal{D}\boldsymbol{y}_i. \tag{71}$$

We are not interested in $\boldsymbol{p}_{i|\sigma_c}^*$ because it is computed for the wrong delay and only on a small subset of the training data. However, by the above assumptions, it is enough to find good (likely not optimal) values for the supports. Thus we only care about the norm of the residual with respect to the optimal weights $\boldsymbol{r}_{i|\sigma_c}^*$. A quick calculation reveals

$$\left\| \boldsymbol{r}_{i|\sigma_c}^* \right\|_2^2 = \left\| \boldsymbol{A}(\sigma^c)\boldsymbol{p}_{i|\sigma_c}^* - \mathcal{D}\boldsymbol{y}_i \right\|_2^2 = \left( \boldsymbol{A}(\sigma^c)\boldsymbol{p}_{i|\sigma_c}^* - \mathcal{D}\boldsymbol{y}_i \right)^\top \left( \boldsymbol{A}(\sigma^c)\boldsymbol{p}_{i|\sigma_c}^* - \mathcal{D}\boldsymbol{y}_i \right) \tag{72}$$

$$= \boldsymbol{p}_{i|\sigma_c}^{*\top} \boldsymbol{A}(\sigma^c)^\top \boldsymbol{A}(\sigma^c)\boldsymbol{p}_{i|\sigma_c}^* - 2\mathcal{D}\boldsymbol{y}_i^\top \boldsymbol{A}(\sigma^c)\boldsymbol{p}_{i|\sigma_c}^* + \mathcal{D}\boldsymbol{y}_i^\top \mathcal{D}\boldsymbol{y}_i \tag{73}$$

$$= \mathcal{D}\boldsymbol{y}_i^\top \boldsymbol{Q}_{\sigma^c} \boldsymbol{R}_{\sigma^c}^{-\top} \boldsymbol{R}_{\sigma^c}^\top \boldsymbol{Q}_{\sigma^c}^\top \boldsymbol{Q}_{\sigma^c} \boldsymbol{R}_{\sigma^c} \boldsymbol{R}_{\sigma^c}^{-1} \boldsymbol{Q}_{\sigma^c}^\top \mathcal{D}\boldsymbol{y}_i - 2\mathcal{D}\boldsymbol{y}_i^\top \boldsymbol{Q}_{\sigma^c} \boldsymbol{R}_{\sigma^c} \boldsymbol{R}_{\sigma^c}^{-1} \boldsymbol{Q}_{\sigma^c}^\top \mathcal{D}\boldsymbol{y}_i + \mathcal{D}\boldsymbol{y}_i^\top \mathcal{D}\boldsymbol{y}_i \tag{74}$$

$$= \left\| \mathcal{D}\boldsymbol{y}_i \right\|_2^2 - \left\| \boldsymbol{Q}_{\sigma^c}^\top \mathcal{D}\boldsymbol{y}_i \right\|_2^2. \tag{75}$$

**Cost Analysis.** Thus, evaluating $N_{\sigma_c}^{\mathrm{O}}$ candidates requires $N_{L+1}$ target vector norms, $N_{\sigma^c}^{\mathrm{O}}$ matrix assemblies and thin $QR$ decompositions, and $N_{L+1}N_{\sigma^c}^{\mathrm{O}}$ $\boldsymbol{Q}$-$\boldsymbol{y}$ products and norms. This cost is neglegible compared to the following weight computation since the expensive operations, namely the matrix assemblies and decompositions, only scale in the number of candidates.

**Choice of Candidates.** Using assumption (b), the number of candidates can be kept small by a proper choice of spacing. Concretely, we propose to use a power-law spacing with exponent $\alpha$ between a small $\sigma_{\min}^{\mathrm{O}}$ and large $\sigma_{\min}^{\mathrm{O}}$ with respect to $H$, given by

$$P_\alpha(a, b, N) = \left\{ \left( \left(1 - \frac{m-1}{N}\right) a^{1/\alpha} + \frac{m-1}{N} b^{1/\alpha} \right)^\alpha \right\}_{m=1}^N, \tag{76}$$

i.e. linearly spaced between $a^{1/\alpha}$ and $b^{1/\alpha}$ for a moderate $\alpha \in (1, \infty)$ as a compromise between linear ($\alpha = 1$) and logarithmic ($\alpha \to \infty$) spacing, since linear spacing violates (b) and logarithmic spacing clusters too aggressively around small values. We suggest using $\sigma_{\min}^{\mathrm{O}} = 1$ and $\sigma_{\max}^{\mathrm{O}} = 2H$ timesteps, $\alpha = 1.5$, and $N_{\sigma^c}^{\mathrm{O}} = 30$ as reasonable defaults, which should require no further tuning for most datasets.

**Summary.** Compactly, the substep $\Sigma_{\sigma_{\min}^{\mathrm{o}}, \sigma_{\max}^{\mathrm{o}}, \alpha, N_{\sigma^c}^{\mathrm{o}}}$ is given by

$$
\Sigma_{\sigma_{\min}^{\mathrm{o}}, \sigma_{\max}^{\mathrm{o}}, \alpha, N_{\sigma^c}^{\mathrm{o}}} \left( \left\{ \Psi^{(L)} \left( x_I^{(n)} \right) \right\}_{n=1}^{\tilde{M}}, \left\{ y_I^{(n)} \right\}_{n=1}^{\tilde{M}} \right) =
$$

$$
\left( \operatorname*{arg\,min}_{\sigma_c \in P_\alpha \left( \sigma_{\min}^{\mathrm{o}}, \sigma_{\max}^{\mathrm{o}}, N_{\sigma^c}^{\mathrm{o}} \right)} \left\| \begin{pmatrix} \mathcal{D}y_{I,i}^{(1)} \\ \mathcal{D}y_{I,i}^{(2)} \\ \vdots \\ \mathcal{D}y_{I,i}^{(\tilde{M})} \end{pmatrix} \right\|_2^2 - \left\| \boldsymbol{Q}_{\sigma^c}^\top \begin{pmatrix} \mathcal{D}y_{I,i}^{(1)} \\ \mathcal{D}y_{I,i}^{(2)} \\ \vdots \\ \mathcal{D}y_{I,i}^{(\tilde{M})} \end{pmatrix} \right\|_2^2 \right)_{i=1}^{N_{L+1}}
$$

(77)

with $\boldsymbol{Q}_{\sigma^c} \boldsymbol{R}_{\sigma^c} = \boldsymbol{A}(\sigma^c)$ as in equation 68 using the same discretisation operator $\mathcal{D}$ as for $y$.

While this approach is fast and works reasonably well, it is far from elegant and constitutes a major avenue for improvement in future iterations of the algorithm.

## A.7 OUTPUT WEIGHTS

**Derivation of the linear system.** We will first derive the idea for a general case before applying it to the problem at hand. For the moment consider the general problem over some Hilbert Space $\mathcal{H}$ equipped with the inner product $\langle \cdot, \cdot \rangle_{\mathcal{H}}$ of best approximating a finite set of target vectors $\left\{ f^{(n)} \right\}_{n=1}^N$ using a real linear combination of a finite set of sample dependent ansatz vectors $\left\{ \left\{ f_k^{(n)} \right\}_{k=1}^K \right\}_{n=1}^N$ with respect to the metric induced by $\langle \cdot, \cdot \rangle_{\mathcal{H}}$ and an $\ell^2$-regularisation term with weight $\lambda$ applied on the norm of the coefficents. Formally, the problem is stated as

$$
\min_{w \in \mathbb{R}^K} \sum_{n=1}^N \left\langle f^{(n)} - \sum_{k=1}^K w_k f_k^{(n)}, f^{(n)} - \sum_{k=1}^K w_k f_k^{(n)} \right\rangle_{\mathcal{H}} + \lambda w^\top w.
$$

(78)

It can easily be verified that this is a strictly convex problem in $w$ for $\lambda > 0$ and thus can be approached using first order optimality. Using bilinearity, we get

$$
\frac{\partial}{\partial w_k} \sum_{n=1}^N \left\langle f^{(n)} - \sum_{k_1=1}^K w_k f_{k_1}^{(n)}, f^{(n)} - \sum_{k_1=1}^K w_k f_{k_1}^{(n)} \right\rangle_{\mathcal{H}} + \lambda w^\top w
$$

(79)

$$
= \frac{\partial}{\partial w_k} \sum_{n=1}^N \left\langle f^{(n)}, f^{(n)} \right\rangle_{\mathcal{H}} - 2 \sum_{k=1}^K \left\langle f_k^{(n)}, f^{(n)} \right\rangle_{\mathcal{H}} + \sum_{k_1=1}^K \sum_{k_2=1}^K \left\langle f_{k_1}^{(n)}, f_{k_2}^{(n)} \right\rangle_{\mathcal{H}} + \lambda w^\top w
$$

(80)

$$
= \sum_{n=1}^N -2 \sum_{k=1}^K \left\langle f_k^{(n)}, f^{(n)} \right\rangle_{\mathcal{H}} + 2 \sum_{k_1=1}^K w_{k_1} \left\langle f_{k_1}^{(n)}, f_k^{(n)} \right\rangle_{\mathcal{H}} + 2\lambda \sum_{k_1} w_k = 0
$$

(81)

$$
\iff \sum_{n=1}^N \sum_{k_1=1}^K w_{k_1} \left\langle f_{k_1}^{(n)}, f_k^{(n)} \right\rangle_{\mathcal{H}} + \lambda \sum_{k_1} w_k = \sum_{n=1}^N \sum_{k=1}^K \left\langle f_k^{(n)}, f^{(n)} \right\rangle_{\mathcal{H}},
$$

(82)

which can be written in matrix form using $F_{ij}^{(n)} = \left\langle f_i^{(n)}, f_j^{(n)} \right\rangle_{\mathcal{H}}$ as

$$
Fw = \sum_{n=1}^{N} \begin{pmatrix} F_{11}^{(n)} + \lambda & F_{12}^{(n)} & \cdots & F_{1K}^{(n)} \\ F_{21}^{(n)} & F_{22}^{(n)} + \lambda & \cdots & F_{2K}^{(n)} \\ \vdots & \vdots & \ddots & \vdots \\ F_{K1}^{(n)} & F_{K2}^{(n)} & \cdots & F_{KK}^{(n)} + \lambda \end{pmatrix} \begin{pmatrix} w_1 \\ w_2 \\ \vdots \\ w_K \end{pmatrix} = \sum_{n=1}^{N} \begin{pmatrix} \langle f_1^{(n)}, f^{(n)} \rangle_{\mathcal{H}} \\ \langle f_2^{(n)}, f^{(n)} \rangle_{\mathcal{H}} \\ \vdots \\ \langle f_K^{(n)}, f^{(n)} \rangle_{\mathcal{H}} \end{pmatrix}. \tag{83}
$$

After replacing the vectors $\mathcal{H} \ni \phi \approx \mathcal{D}\phi \in \mathbb{R}^{H_D}$ and inner product $\langle f_1, f_2 \rangle \approx \mathcal{D}f_1^\top \mathcal{D}f_2$ by discrete approximations, the system becomes

$$
\left( \mathcal{D}F^\top \mathcal{D}F + n\lambda \mathrm{Id}_K \right) w = \mathcal{D}F^\top \mathcal{D}f, \tag{84}
$$

which are exactly the regularised normal equations for the matrix

$$
\mathcal{D}F = \begin{bmatrix} \mathcal{D}f_1^{(1)} & \mathcal{D}f_2^{(1)} & \cdots & \mathcal{D}f_K^{(1)} \\ \mathcal{D}f_1^{(2)} & \mathcal{D}f_2^{(2)} & \cdots & \mathcal{D}f_K^{(2)} \\ \vdots & \vdots & \ddots & \vdots \\ \mathcal{D}f_1^{(N)} & \mathcal{D}f_2^{(N)} & \cdots & \mathcal{D}f_K^{(N)} \end{bmatrix} \in \mathbb{R}^{N \cdot H_D \times K} \text{ and vector } \mathcal{D}f = \begin{bmatrix} \mathcal{D}f^{(1)} \\ \mathcal{D}f^{(2)} \\ \vdots \\ \mathcal{D}f^{(N)} \end{bmatrix} \in \mathbb{R}^{N \cdot H_D} \tag{85}
$$

stacked as in equation 68.

Returning to the problem at hand, this digression provides (a) a justification for treating treating the discretised time points as additional samples (under the assumption of equal weighting) and (b) gives a formulation of the linear problem that can be assembled in batches and requires only a small ($N_L + 1 \times N_L + 1$) system per neuron to be solved. Concretely, for neuron $i$, we accumulate the sum in equation 83 in batches by only assembling a moderate amount of rows in $\mathcal{D}F$ in equation 85 before performing the multiplications in equation 84 on the submatrix and the respective rows of the vector $\mathcal{D}F$ with the ansatz functions $\left\{ \{1\} \cup \left\{ \psi_{ij}^{(L+1)} \left( x^{(n)} \right) \right\}_{j=1}^{N_L} \right\}_{n=1}^{M}$ and target vectors $\left\{ y_i^{(n)} \right\}_{n=1}^{M}$. In the case of spike-valued outputs, $\eta_i^{(L+1)}$ evaluated on the target spike trains would be added as an additional ansatz function to solve for the spike cost.

**Conditioning of the System Matrix.** We provide a brief estimation of the condition number of the resulting matrix

$$
F = \sum_{n=1}^{M} \mathcal{D}F^{(n)\top} \mathcal{D}F^{(n)} + M\lambda \mathrm{Id}_{N_L+1} := F_S + M\lambda \mathrm{Id}_{N_L+1} \tag{86}
$$

summed over the entire training set. Each summand $\mathcal{D}F^{(n)\top} \mathcal{D}F^{(n)} + \lambda \mathrm{Id}_{N_L+1}$ is symmetric and (strictly) positive definite for $\lambda > 0$. An immediate loose bound

$$
\kappa(F) = \frac{\lambda_{\max}(F)}{\lambda_{\min}(F)} = \frac{M\lambda + \lambda_{\max}(F_S)}{M\lambda + \lambda_{\min}(F_S)} \leq \frac{M\lambda + \sum_{n=1}^{M} \lambda_{\max}(\mathcal{D}F^{(n)\top} \mathcal{D}F^{(n)})}{M\lambda} \tag{87}
$$

is provided by Weyl's inequality, where it can be seen that the regularisation immediately guaruantees finity. Bounding the spectral norm by the Frobenius norm through $\left\| \mathcal{D}F^{(n)} \right\|_2^2 \leq \left\| \mathcal{D}F^{(n)} \right\|_F^2 = \sum_{j=1}^{N_L+1} \left\| \mathcal{D}F_{:j}^{(n)} \right\|_2^2$ and plugging in the specific ansatz functions, we get

$$
\kappa(F) \leq \frac{M\lambda + \sum_{n=1}^{M} \sum_{j=1}^{N_L+1} \left\| \mathcal{D}F_{:j}^{(n)} \right\|_2^2}{M\lambda} \leq 1 + \frac{H_D}{\lambda} + \frac{\left\| \mathcal{D}k_i^{(L+1)} \right\|_2^2}{M\lambda} \sum_{n=1}^{M} \sum_{j=1}^{N_L} \left| T_j^{(L)} \right|^2, \tag{88}
$$

relating the condition to the spike counts $\left|T_j^{(L)}\right|$, since the design matrix contains a column of 1s for the bias and each spike contributes (at most) a full copy of the discretised kernel $\mathcal{D}k_i^{(L+1)}$, where $\mathcal{D}F_{:j}^{(n)}$ is the $j$-th column of $\mathcal{D}F_{:j}^{(n)}$ and

$$
k_i^{(L+1)}(t) = k^{(L+1)}\left.\left(\frac{t - \tau_i^{(L+1)}}{\sigma_i^{(L+1)}}\right)\right|_{t \geq 0}. \tag{89}
$$

Since the neurons in the last hidden layer will typically code for different features by having different temporal parameters and the separation criterion, both the Weyl's and Frobenius estimations are rather conservative here. Nonetheless, this bound yields several useful insights. Firstly, considering for example, $\lambda \geq 10^{-3}$, $N_L$ and $\left|T_j^{(L)}\right|^2 \in \mathcal{O}(10^2)$ and $\left\|\mathcal{D}k_i^{(L+1)}\right\|_2^2 \in \mathcal{O}(10)$, we find that $\kappa(F) \in \mathcal{O}(10^8)$ as a loose upper bound, suggesting that while the system is unlikely to be *catastrophically* ill-conditioned, the possibility of ill-conditioning can not be neglected. Secondly, the benefits of a sparse spiking representation can go beyond mere efficiency, which is intuitive in this case, since more spikes enable more complicated functions to be represented. Finally, the price for "compressing" the time dimension in the linear problem is paid in the condition, both through the direct $H_D$ contribution and the number of spikes, since longer evaluation intervals will typically mean more contributing spikes.

These estimations are in line with the findings of our numerical experiments that the condition rarely caused immediately obvious issues, which is why no preconditioning was applied during the experiments described in the following section. Future work should, however, investigate the relevance of bad conditioning to the algorithm and possible mitigationg strategies.

**Regularisation Parameter Search.** Finally, since the matrix $F_S$ is symmetric and applying the diagonal regularisation changes only the eigenvalues but not the eigenvectors, it is possible to search over a set of candidate regularisation terms with little added effort. Using the pre-assembled system matrix $F_S$ and the target vector $y = \mathcal{D}F^\top \mathcal{D}f$ for the training set and assembling the respective system matrix on a validation subset, computing its symmetric eigendecomposition $F_S = \Gamma \Lambda \Gamma^\top$ allows to cheaply test a candidate regularisation $\lambda^c$ without re-solving and re-assembling the full system by using that the optimal parameters are given as $p_{\lambda^c}^* = \Gamma(\Lambda + \lambda^c \mathrm{Id}_{N_L+1})^{-1}\Gamma^\top y$, which can then be evaluated on the validation set by using the pre-assembled system matrix. In *S-SWIM*, we use this strategy and test 32 logarithmically spaced candidates in $10^{-5} \leq \lambda^c \leq \frac{1}{2}$. It should be emphasised that this does not mean computing the weights from the validation set. The specific formulation employed simply allows moving an outer optimisation loop over the regularisation parameter into the solution of the linear problem. The eigendecomposition in this step did not converge in some experiments during the ablation study discussed in section C.2 (Ablation Study). It remains to be answered whether this is related to an ill-conditioned system matrix.

## B EXPERIMENTAL SETTINGS

To ensure reproducibility, the full details of the conducted experiments are given in the following.

### B.1 IMPLEMENTATION & SETUP

We implement algorithm 1 in the Python framework *CuPy* (Okuta et al., 2017). All experiments were conducted on a system equipped with an AMD EPYC 7402 processor (2.80 GHz, single-socket, 24 cores per socket with 2-way hyper-threading), 256 GiB of system memory, and four NVIDIA RTX 3080 Turbo GPUs, each with 10 GiB of video memory. Each Experiment only used a single GPU. All reported results are averaged across three seeds.

### B.2 TIME-SERIES FORECASTING

#### B.2.1 DATASETS & EVALUATION METRICS

Following Lv et al. (2025), we choose two short- and two long-term observation length datasets to evaluate the model. The characteristics of each dataset are summarised in Table 2.

Table 2: The statistics of time-series datasets. Reproduced from (Lv et al., 2025, Table 4).

| Dataset | Samples | Variables | Observation Length | Train-Valid-Test Ratio |
|---|---|---|---|---|
| Metr-la | $34,272$ | $207$ | $12$, (short-term) | $(0.7, 0.2, 0.1)$ |
| Pems-bay | $52,116$ | $325$ | $12$, (short-term) | $(0.7, 0.2, 0.1)$ |
| Solar-energy | $52,560$ | $137$ | $168$, (long-term) | $(0.6, 0.2, 0.2)$ |
| Electricity | $26,304$ | $321$ | $168$, (long-term) | $(0.6, 0.2, 0.2)$ |

As evaluation metric, we use the Root Relative Squared Error (*RSE*)

$$
\text{RSE} = \sqrt{\frac{\sum_{m=1}^{M} \left\| \boldsymbol{Y}^{(m)} - \hat{\boldsymbol{Y}}^{(m)} \right\|_F^2}{\sum_{m=1}^{M} \left\| \boldsymbol{Y}^{(m)} - \bar{\boldsymbol{Y}} \right\|_F^2}}.
\tag{90}
$$

Here, $\mathbb{R}^{C \times L} \ni \boldsymbol{Y}^{(m)} = \left( Y_{c,l}^{(m)} \right)_{c=1,\dots C; l=1,\dots L}$, $Y_{c,l}^{(m)}$ denotes the $l$-th target value of the $c$-th variable in the $m$-th sample of the evaluation set and $\hat{Y}_{c,l}^{(m)}$ the respective model prediction. $\bar{\{\cdot\}}$ denotes averaging over all samples of the evaluation set. All datasets were normalised to lie in the range $[0, 1]$ for all experiments.

#### B.2.2 SURROGATE-GRADIENT TRAINING

As reference, the same models were trained with *SGD* using a surrogate-gradient strategy. Specifically, we used the Lava DL (Intel Neuromorphic Computing Lab & lava-nc contributors, 2024) implementation of the *SLAYER* (Shrestha & Orchard, 2018) algorithm. *SLAYER* was chosen because it was built for the same delay-parameterisation used in definition 3.1. Optimisation was performed with respect to **all** parameters in the proposed model.

To initialise weights for *SGD*-only training, we use an adaptation of the strategy proposed in Rossbroich et al. (2022) to *SRM* neuron with $\mu = 0.5$ and $\xi = 1$. The biases were either initialised to zero if the weight

Table 3: Hyperparameters of gradient-based training for *SGD*-only case and the fine-tuning starting from a *S-SWIM* trained network (Time-Series Forecasting).

| Parameter | Value for *SGD*-only | Value for fine-tuning |
|---|---|---|
| Batch Size | 64 | 64 |
| Learning Rate Schedule | Cosine schedule | Cosine schedule |
| Initial Learning Rate | $5 \cdot 10^{-4}$ | $2.5 \cdot 10^{-4}$ |
| Final Learning Rate | $1 \cdot 10^{-5}$ | $1 \cdot 10^{-5}$ |
| Maximum Nr. of Epochs | 750 | 250 |
| $\lambda_{\text{reg}}$ | $10^{-4}$ | From *S-SWIM* |
| Early Stopping Patience | 30 | 30 |
| Early Stopping Min. Improvment | $10^{-6}$ | $10^{-6}$ |

distribution was well defined, or by $b = 2\left(\left(\frac{\xi\sqrt{\hat{\epsilon}}}{\epsilon\sqrt{n\nu}} + 1\right)^{-1} - \mu\right)$ in the notation of Rossbroich et al. (2022) to enforce a strictly positive standard deviation, otherwise. The mean input $\nu$ was computed over a subbatch of 1000 samples. The hidden layer delays were initialised by $\tau \sim \text{Uniform}(0, 15)$, the kernel supports by $\sigma \sim \text{Uniform}(5, 15)$ timesteps. Spike costs were initialised to $c = 1$. The output layer delays were initialised as $\tau = 0$, whereas the *SRK* supports were set to $\sigma = 15$ timesteps. Mean square error (*MSE*) was used as loss function with regularisation on the 2-norm of the output layers' weights weighted by $\lambda_{\text{reg}}$.

### B.2.3 MODEL ARCHITECTURE & HYPERPARAMETERS

The model was composed of one spiking hidden layer with 750 neurons and one non-spiking output layer whose voltage was taken as the model prediction ($L = 1$ in definition 3.1). For the hidden layer, either a hat function

$$\text{Hat}(x) = \max(1 - |x|, 0) \tag{91}$$

or a rectified (rescaled) Morlet

$$\text{Morlet}(x) = \begin{cases} \exp\left(-3x^2\right)\cos\left(2\pi x\right) & \text{for } |x| \leq 1, \\ 0 & \text{otherwise} \end{cases} \tag{92}$$

was used as *SRK*, while the *RfK* of the hidden layer was always a rectified decaying exponential

$$q(x) = \begin{cases} \exp\left(-x\right) & \text{for } |x| \leq 1, \\ 0 & \text{otherwise.} \end{cases} \tag{93}$$

The rectification was applied to simplify the computation of convolutions.

The hyperparameters used for *SGD*-only training and fine-tuning are given in Table 3. The hyperparameters used in the *S-SWIM* algorithm are given in Table 4.

### B.3 CLASSIFICATION

#### B.3.1 DATASETS & PRE-PROCESSING

To show experiments on both spiking and non-spiking classification datasets, we choose two static image datasets (MNSIT and *F-MNIST*) and the well known speech recognition Spiking Heidelberg Digits (*SHD*). The datasets are summarised in Table 5.

Table 4: Hyperparameters for *S-SWIM* training (Time-Series Forecasting).

| Substep | Parameter | Value |
|---|---|---|
| Full Algorithm | $\tilde{M}$ | 1000 |
| Temporal Parameters $\mathcal{T}$ | $\tau_{\max}$ | $\begin{cases} H & \text{if } H > O, \\ O/2 & \text{otherwise} \end{cases}$ |
| | $\sigma_{\min}^h$ | 5 |
| | $\sigma_{\max}^h$ | 50 |
| | $N_\sigma^h$ | 10 |
| Sampling Distribution | Metrics | By Entropy |
| Sampled Weights $\mathcal{W}$ | Criterion | Dot |
| Normaliser $\mathcal{Z}$ | Criterion | *MS* |
| | $\mu_t$ | 0.5 |
| | $s_t$ | 0.5 |

Table 5: Overview of classification datasets. Non-spiking datasets are transformed to a spiking representation for the experiments. The original 700 input channels of *SHD* are reduced to 70 by merging neighbouring channels.

| Dataset | Source | Input Variables | Classes | Spiking | Train/Valid/Test Samples |
|---|---|---|---|---|---|
| MNIST | Deng (2012) | 784 | 10 | No | 50K/10K/10K |
| Fashion-MNIST (F-MNIST) | Xiao et al. (2017) | 784 | 10 | No | 50K/10K/10K |
| SHD | Cramer et al. (2020) | 70 (700) | 20 | Yes | 8156/-/2264 |

**Images:** To transform images into spike trains, we follow Dai & Ma (2025) and sample a number of spikes for each pixel in the flattened images from a Poisson distribution with rate

$$\lambda_i^{(n)} = \alpha \cdot O \cdot \frac{p_i^{(n)} - p_{\min}}{p_{\max} - p_{\min}}, \tag{94}$$

where $p_{\min} \leq p_i^{(n)} \leq p_{\max}$ is the intensity of pixel $i$ in the flattened image $n$ and $p_{\min}, p_{\max}$ respectively denote the minimum and maximum intensity of the encoding and $\alpha \in (0,1)$. Those spikes are then spread uniformly over the interval $[0, O)$ and presented as the input to the network. For our experiments, we set $\alpha = 0.2$, $O = 100$ and evaluate the model with a timestep of 1. Thus, black pixels correspond to (in expectation) no spikes, whereas white pixels correspond to one spike every 5 time steps.

**SHD:** The *SHD* test set - by design - contains speakers that were not present in the training set, so there is a significant distribution shift (Cramer et al. (2020)). To counteract this, other works apply substantial preprocessing such as filtering and binning spikes and merging channels (see e.g. Yin et al. (2020)) effectively undoing the distribution shift by removing high-frequency information. Since the purpose of this study is to show the applicability of our method rather than achieve state-of-the-art performance, we only perform the channel merging as in Cramer et al. (2020), as this also reduces the memory footprint of the dataset. The spike-times are not binned but fed into the model with their exact times. Rather than choosing the timestep as $0.01$, we rescale all spike times by a factor of 100 and evaluate with a timestep of 1, respectively setting $O = 140$, as the latest spike in the dataset occurs around 1.35 seconds before rescaling.

### B.3.2 Readout & Surrogate Voltage

As usual, we set the number of neurons in the last layer to the number of classes. The predicted class $\hat{K}$ is then

$$\hat{K} = \arg\max_i \int_0^{T+H} \Psi_i^{(L+1)}(\boldsymbol{x}, t)\, \mathrm{d}t, \tag{95}$$

which is the most positive voltage or the number of spikes depending on how the final layer is set up (cf. definition 3.1 and Remark 3.2). This motivates the construction of the surrogate voltage used as target for the regression problem by

$$y_i^{surr}(t) = \begin{cases} (1+\Delta)\theta & \text{for } i = K, \\ -(1+\Delta)\theta & \text{otherwise,} \end{cases} \tag{96}$$

where $K$ is the ground-truth class and $\Delta > 0$. This directly enforces the most positive voltage and is a reasonable stand-in objective for the most spikes. The exact choice of $\Delta$ matters little as multiplicative constants are directly absorbed into the least-squares solution (cf. section A.7). In our experiments, we set $\Delta = 1$ for simplicity. The prediction horizon $H$ was set to $15$ time steps for all experiments. As the employed criterion is invariant (up to finite-window effects) to temporal shifts, we do not fit the delays of the last layer, but simply set them to $0$.

It should be emphasised that there is no good reason to perform the readout on static classification problems using spikes. The discussion here solely serves to illustrate the idea of surrogate voltages.

### B.3.3 Network Architecture & Hyperparameters

We test models with one or two spiking hidden layers ($L \in \{1, 2\}$ in definition 3.1) and a varying number of neurons in the hidden layers, followed by a spiking layer from which we compute the predicted class either by integrating the total voltage or counting the number of spikes (cf. section B.3.2). For the first hidden layer, a hat function

$$\text{Hat}(x) = \max(1 - |x|, 0) \tag{97}$$

was used for the image benchmarks and a rectified Morlet

$$\text{Morlet}(x) = \begin{cases} \exp\left(-x^2\right)\cos\left(2\pi x\right) & \text{for } |x| \le 1, \\ 0 & \text{otherwise} \end{cases} \tag{98}$$

was used as *SRK* for *SHD* and Hat for all following layers in both cases. The *RfK* of all layers was always a rectified decaying exponential

$$q(x) = \begin{cases} \exp\left(-x\right) & \text{for } |x| \le 1, \\ 0 & \text{otherwise.} \end{cases} \tag{99}$$

The rectification was applied to simplify the computation of convolutions.

For sampling the distance in the output space was defined as $0$ within classes and $1$ between classes. We evaluate a small set of architecture configurations and hyperparameters for both benchmarks, which are found in Table 6. Random weights were sampled from a standard normal distribution; only $\mathcal{W}$ changes, all other steps of the algorithm are still executed as stated.

Table 6: Architecture and Hyperparameters for the Classification Experiments.

| Component | Parameter | Tested Values |
|---|---|---|
| Network Architecture | $N_l$ (all hidden layers) | $\{256, 512, 1000\}$ |
| | $L$ | $\{1, 2\}$ |
| Full Algorithm | $\tilde{M}$ | 900 |
| Temporal Parameters $\mathcal{T}$ | $\tau_{\max}$ | 30 |
| | $\sigma^h_{\min}$ | 3 |
| | $\sigma^h_{\max}$ | 25 |
| | $N^h_\sigma$ | 10 |
| Sampling Distribution | $d_{\mathcal{Y}}$ | Class Distance |
| | $(d_{\mathcal{X}_0}, d_{\mathcal{X}_l})$ | $\{(L^2, L^2),$ (mag, $\angle)\}$ |
| Sampled Weights $\mathcal{W}$ | Criterion | $\{$Dot, Random$\}$ |
| Normaliser $\mathcal{Z}$ | Criterion | $MS$ |
| | $\mu_t$ | 0.5 |
| | $s_t$ | 0.5 |

## C    FURTHER EXPERIMENTS

We investigate the robustness of individual components through further numerical experiments.

### C.1    CLASSIFICATION WITH SPIKING READOUT

Here, we discuss the results of evaluating the predicted class based on the spikes output by the final layer. It should be emphasized that the network architecture and training method are unchanged only the readout is different. This is more of an illustrative example rather than a practical application of this idea. The results are shown in figure 4. We find that the idea of surrogate voltages does work as the performance is consistently above chance level and shows the same patterns as for the voltage readout. However, the performance is significantly lower than for voltages. Thus, perhaps more carefully chosen surrogates are needed. However, it is also likely that measures such as normalisation similar to the hidden layers will counteract this performance drop greatly, which we however did not look into here.

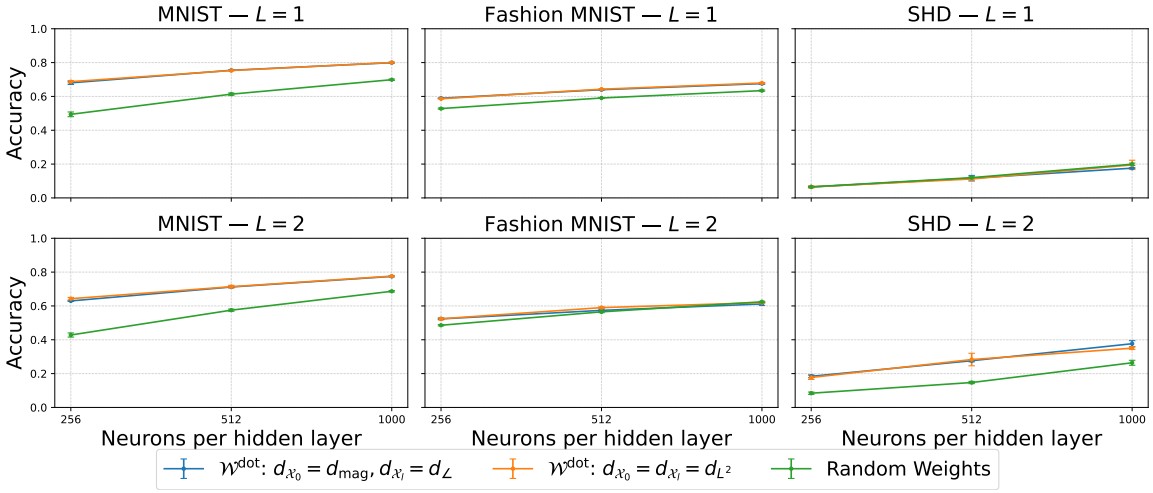

Figure 4: Classification accuracy using spiking readout.

### C.2    ABLATION STUDY

Concretely, we try to answer the questions of whether (a) the proposed weight construction criteria perform better than random sampling and (b) the neurons in the hidden layer are tuned to different features. To this end, we evaluate the performance of *S-SWIM* trained networks using $\mathcal{W}^{\text{dist}}$, $\mathcal{W}^{\text{dot}}$, or drawing weights from a standard normal distribution across different dimensionality of the hidden layer on one short (**Pems-bay**) and one long (**Electricity**) observation length dataset. For comprehensiveness, we perform the experiments with different normalisations on the hidden layer weights.

The results are shown in figure 5. Regarding (a), we find that $\mathcal{W}^{\text{dot}}$ consistently and significantly outperforms $\mathcal{W}^{\text{dist}}$ and random weights, whereas $\mathcal{W}^{\text{dist}}$ sometimes performs better, sometimes worse than random sampling, depending on the dataset and normaliser. Regarding (b), we find that across all combinations, the performance increases with increasing number of neurons, which shows that neurons do indeed encode different features. Regarding normalisers, we note that $\mathcal{N}^{\text{MS}}$ is seemingly more stable across parametes than $\mathcal{N}^{\text{Fl}}$.

We assume that, since there is no interpretable pattern in the out-of-bounds values regarding objectives, normalisers, or number of neurons, it must be caused by numerical issues likely linked to the condition of the linear system.

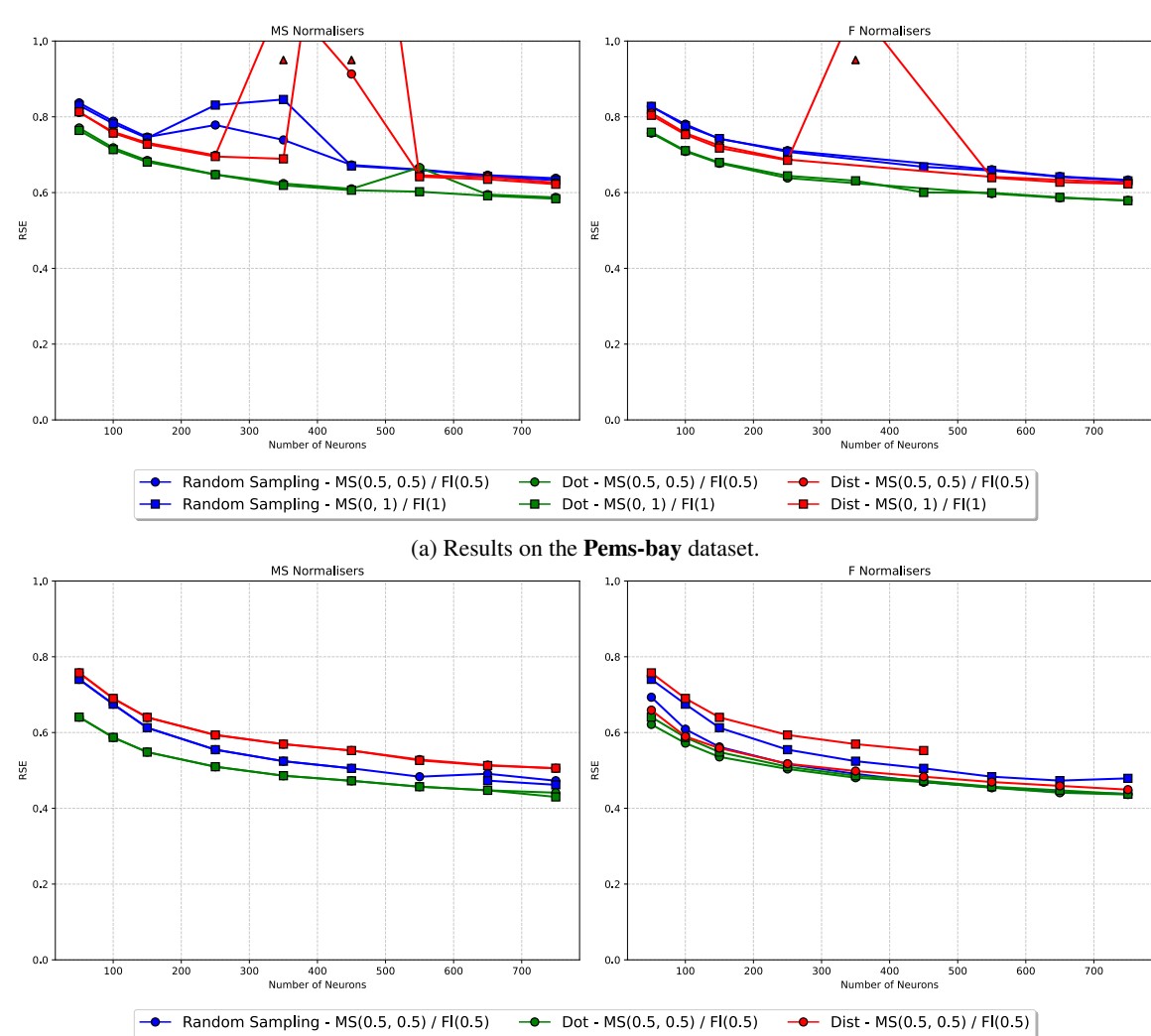

(a) Results on the **Pems-bay** dataset.

(b) Results on the **Electricity** dataset.

Figure 5: Results of the ablation study. Arrows indicate significantly underperforming models. Missing results indicate errors during the computation.

## C.3 ENTROPY CRITERION

Finally, we test how well the proposed metric selection criterion performs compared to prescribing a given pair. We evaluate the performance on all datasets for $H \in \{6, 24, 48\}$ when using the same pseudometric for input and output for each of the proposed feature embeddings, against using the pair that minimises the

entropy. A visualisation of the results is given in figure 6. Firstly, we note that the influence of the used embeddings is, in general, not very large. Concretely, the difference is limited to $0.01 - 0.02$ points of RSE score, which is not insignifant but also not major. This indicates that none of the proposed pseudometrics performs significantly better or worse than the others when using them for both in- and output. It remains to be answered whether a different construction for the sampling metrics would lead to significant improvements. Furthermore, if there is a pair of metrics performing significantly better, the entropy criterion does not select it. Across the given datasets, using the high-frequency band for short horizons and cosine distance for long horizons almost always outperforms the entropy criterion.

In summary, future work should evaluate other constructions than the proposed for evaluating the sampling distribution and derive more robust metric selection criteria.

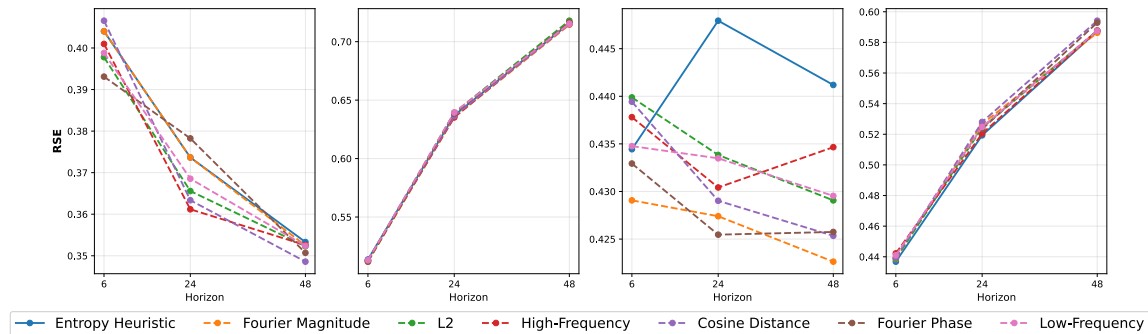

Figure 6: Prediction performance across datasets for different sampling metrics.

## D  LIMITATIONS & FUTURE WORK

**Limitations**   While the performed experiments validate the overall design of the algorithm and show that, given further refinement, *S-SWIM* can become a strong alternative to gradient-based methods, several limitations and much room for improvement remain. Several important parameters are currently still chosen by data-agnostic heuristics. The properties of the sampling distribution and the question of what characteristics of the data are (most) important for sampling require more thorough analysis. Furthermore, only a very limited set of network architectures and hyperparameters was evaluated. Additionally, more research into the numerical stability of the algorithm is needed. Finally, the algorithm was only evaluated on one type of task and currently requires an explicitly prescribed target voltage for the output layer, limiting its applicability to spike-valued datasets.

**Future Work**   We propose the following four points as the major focus of future research. Firstly, more work on sampling the temporal parameters in the hidden layers, especially the kernel parameters, is needed. Future research should look into relating them to characteristic temporal properties of the data, such as autocorrelations or pronounced frequency bands. Secondly, a comprehensive evaluation of possible sampling metrics is needed, especially for sampling multi-layered networks. Thirdly, further work should investigate alternatives to the current search-based approach for the kernel supports in the output layer. We hypothesise that a similarly elegant "identification" approach as for the delays can be constructed. Finally, a method for constructing suitable surrogates in the absence of explicit target voltages is needed. A good start here will likely be deriving the specific set of conditions that make a voltage trace well-approximable by the final layer given the initialisation of the hidden layers.

