# OpenReview forum: "Random Feature Spiking Neural Networks"
_ICLR.cc/2026/Conference — Submitted to ICLR 2026_

### Official Review · Reviewer_k7CL · 2025-10-24

**Soundness:** 2
**Presentation:** 1
**Contribution:** 3
**Rating:** 2
**Confidence:** 2

**Summary:**

This paper proposes S-SWIM, a gradient-free method to train SNNs for time series forecasting. Experimental results have shown its effectiveness.

**Strengths:**

The S-SWIM method is novel for SNN training. Especially, it does not need gradient backpropagation.

**Weaknesses:**

1. This paper is difficult to understand. I recommend the authors to draw an illustration figure of the S-SWIM method to help the readers better understand it.
2. The paper is written like a technical report rather than a paper. For a paper, the authors should propose the scientific problem this paper aims to address, then introduce how the method addresses this problem. The whole part of Section 3 introduces the technical details without holistic intuition. Also the experimental settings (baselines, training method details, etc.) are vague in Section 4.
3. The baselines in Table 1 are not cited. I am not familiar with these works, as far as I know, Spikformer and QKFormer are designed for vision tasks, how do the authors use them in time series forecasting tasks? In addition, it seems that works in 'SNNs for Time Series Forecasting' in Related work are not included in this table.

**Questions:**

1. There lacks the explanation for Eq (1)(2)(3). I recommend the authors to introduce $\phi$ and $\eta$ in Eq (2)(3) (spike response kernel and reset kernel). Besides, what is PSPK, RfKin line 133? A full name should be given. Where do the spike-cost $c$ appear in the equations?
2. In line 157, it seems that $\tau_{max}$ is not continuous at $O=H$. This design is a bit weird, does it have any intuitions?
3. In Eq. (6)(7), what is $-3 \cdot s_t$ in the normal distribution? Normal distributions only have two parameters in common.
4. Authors have said that S-SWIM is much faster than SGD. Where is the experimental data to support this conclusion?

---

> ### Author Response · Authors · 2025-11-19
> **Adressing Weakness 3**
>
> **Regarding the baselines in Table 1:** The results are sourced from [1], as stated in the table description:
>
> *“[…] Results highlighted with shading are ours; the remaining results were sourced from Lv et al. (2025)Table 1.”.*
>
> The results were sourced directly from the named paper, so the details for how those architectures are adapted to time series tasks can be found there. In related work, we mention the works by Lv et al. 24 [2], Lv et al. 25 [1], and Feng et al. 25 [3]. We directly compare to [1] in the Experiments section, which includes results from [1] and [2]. Due to space constraints, we only selected the best-performing models and omitted the results from [3], since the performance was, on average, very similar to [1].
>
> **References:**
> [1] Changze Lv, Yansen Wang, Dongqi Han, Yifei Shen, Xiaoqing Zheng, Xuanjing Huang, and Dongsheng Li. Toward relative positional encoding in spiking transformers, 2025.
>
> [2] Changze Lv, Dongqi Han, Yansen Wang, Xiaoqing Zheng, Xuanjing Huang, and Dongsheng Li. Advancing spiking neural networks for sequential modeling with central pattern generators, 2024.
>
> [3] Shibo Feng, Wanjin Feng, Xingyu Gao, Peilin Zhao, and Zhiqi Shen. Ts-lif: A temporal segment spiking neuron network for time series forecasting, 2025.

---

> ### Author Response · Authors · 2025-11-19
> **Adressing Question 1**
>
> We thankfully acknowledge this feedback and will  improve the readability of this section (update to PDF coming soon). The architecture is a standard linear Spike Response Model Network as discussed in [4] or employed as a model in [5]. The architecture does not constitute a novel contribution of our paper; we merely cast it in a slightly different notation and parameterisation than other works to ease the presentation of our proposed method. Note that when absorbing the Delay and Support parameters into the kernels $k^{(l)}$ and $q^{(l)}$, the model is equivalent to that used in [5] except for the inclusion of a bias parameter. We will make this clearer and include resources providing an explanation of the model when revising this section.
>
> **Regarding $\psi$ and $\eta$ defined in equations (2) and (3)**, they are respectively the contributions from inputs and refractory contributions from the layer’s own spikes. The abbreviations PSPK and RfK were indeed not introduced before use, which we will correct. PSPK refers to $k^{(l)}$ and stands for Post-Synaptic-Potential-Kernel, whereas RfK refers to $q^{(l)}$ and stands for Refractory-Kernel. We chose Post-Synaptic rather than Spike-Response since we allow the inputs to the first layer, which the kernels are convolved with, to be real- or spike-valued.
>
> **The Spike-Costs** $c$ weigh the refractory contribution of $\eta_i^{(l)}$ in equation (1) in the $0<l\leq L$ case.
>
> **References:**
> [4] Wulfram Gerstner, Werner M. Kistler, Richard Naud, and Liam Paninski. Neuronal Dynamics: From Single Neurons to Networks and Models of Cognition. Cambridge University Press, USA, 2014. ISBN 1107635195.
>
> [5] Sumit Bam Shrestha and Garrick Orchard. Slayer: Spike layer error reassignment in time, 2018.

---

> ### Author Response · Authors · 2025-11-19
> **Adressing Question 2**
>
> This is a good observation. We had indeed not considered this edge case. The given choices were obtained empirically. We will update this in the paper and properly mark $\tau_{\max}$ as a hyperparameter to be tuned manually or through search.
>
> The values arose from the intuition outlined in Section 3.2.1 Temporal Parameters. The delays serve two purposes, namely (a) ensuring that a large enough interval containing spikes is available to downstream layers and (b) information sufficiently far into the past is accessible at a given time point. The given construction arose heuristically from the considered datasets, which contained two $O>>H$, where we found $\tau_{\max}=O/2$ to be a reasonable choice regarding (b), and two $O<<H$ cases, where we chose $\tau_{\max}=H$ to ensure (a). As described in Appendix D, in the future we will look into replacing this heuristic / manual choice by a data-driven one, e.g. by considering characteristic scales in the autocorrelation of the input data.

---

> ### Author Response · Authors · 2025-11-19
> **Adressing Question 3**
>
> As stated at the top of Algorithm 1 on page 5, $\mathcal{N}$ denotes the employed “Normaliser” here rather than the normal distribution. It is a function mapping the candidate weight directions and data onto the appropriate scaling factor, bias and spike cost to achieve a prescribed set of statistics on the training data.
> We gave this substep a name rather than writing it out in the algorithm since it can easily be swapped out for other approaches more suitable for the given task and data by the modular design of the algorithm.
> Since this symbol is indeed closely associated with the normal distribution, we will change it in the revision.
>
> Edit: we now use $\mathcal{Z}$

---

> ### Author Response · Authors · 2025-11-19
> **Adressing Question 4**
>
> See Figure 1 on Page 9. Note that the times are plotted on log-scale.

---

> ### Author Response · Authors · 2025-11-19
> **Adressing Weakness 1**
>
> This is an excellent suggestion for improving the readability which we will incorporate during the revision.

---

> ### Author Response · Authors · 2025-11-19
> **Adressing Weakness 2**
>
> **Regarding clarity,** we will try to incorporate the suggestions by respectively devoting more space to introducing the problem in Section 1 and the overall approach in Section 3. We will point out the revisions once this is done.
>
>
> **Regarding the experimental settings,** as pointed out in the text, please refer to Appendix B for the details.

---

> ### Comment · Reviewer_k7CL · 2025-11-25
>
> Thank you for your response. This work seems novel and technically solid, but the writing is really hard to follow. Wait for your revised paper.
>
> For Weakness 3, I realized the baselines are from the same work after your response, but I recommend that you emphasize this, since it is confusing if not noticed “[…] Results highlighted with shading are ours; the remaining results were sourced from Lv et al. (2025)Table 1.”

---

### Official Review · Reviewer_w4Dg · 2025-11-02

**Soundness:** 2
**Presentation:** 2
**Contribution:** 2
**Rating:** 4
**Confidence:** 3

**Summary:**

This paper proposes a Random Feature Spiking (RFS) method to enable efficient supervised learning in Spiking Neural Networks (SNNs) by treating membrane dynamics as random projections. The authors offer theoretical guarantees showing consistency to SGD-based weight updates and demonstrate preliminary experimental evaluation.

**Strengths:**

1. The paper provides sufficient theoretical justification for the proposed method.
2. A promising solution for enabling global weight updates in SNNs similar to SGD.

**Weaknesses:**

1. Since the paper states conceptual relationships to SWIM, but does not empirically or algorithmically contrast against it, it remains unclear that whether temporal parameters meaningfully impact learning quality, and whether RFS is fundamentally more scalable or simply a variant of existing methods.
2. Experimental evaluation is insufficient to validate claims
The paper contains only one dataset and one architecture configuration, with missing details such as:
number of synapses / learned parameters in neurons,
Spike-based computation metrics (SynOps / NeuOps).
3. Writing clarity and notation consistency need improvement
Numerous grammatical mistakes, unclear expressions, and inconsistent mathematical notation negatively affect readability. Some expressions lack necessary subscripts or arguments, making it harder to follow derivations.
4. Presentation misses a comprehensive contextual comparison
The related-work section briefly lists methods but does not sufficiently articulate technical distinctions from:
Local learning rules like e-prop [1]
Sparse BP [2,3]
Current efficient training methods like gradient estimation [4] and integer spikes [5]
5. Conclusion overstates findings relative to evidence
The claims of “better credit assignment” remain theoretical without adequate practical demonstration.

[1] Bellec, G., Scherr, F., Subramoney, A. et al. A solution to the learning dilemma for recurrent networks of spiking neurons. Nat Commun 11, 3625 (2020).
[2] Meng, Qingyan, et al. "Towards memory-and time-efficient backpropagation for training spiking neural networks." Proceedings of the IEEE/CVF international conference on computer vision. 2023.
[3] Perez-Nieves, Nicolas, and Dan Goodman. "Sparse spiking gradient descent." Advances in Neural Information Processing Systems 34 (2021): 11795-11808.
[4] Meng, Qingyan, et al. "Training high-performance low-latency spiking neural networks by differentiation on spike representation." Proceedings of the IEEE/CVF conference on computer vision and pattern recognition. 2022.
[5] Luo, Xinhao, et al. "Integer-valued training and spike-driven inference spiking neural network for high-performance and energy-efficient object detection." European Conference on Computer Vision. Cham: Springer Nature Switzerland, 2024.

**Questions:**

see in weakness

---

> ### Author Response · Authors · 2025-11-19
> **Adressing Weakness 1**
>
> SWIM is a method for training classical Multi-Layer-Perceptron networks (MLP) and is not applicable to the spiking neural networks we train here.
> SWIM is useful for non-temporal data, whereas the proposed S-SWIM method applies to SNNs handling spatio-temporal data.

---

> ### Author Response · Authors · 2025-11-19
> **Adressing Weakness 2**
>
> **Regarding experimental evaluation:** We use the same 4 datasets as other recent SNN TSF papers, such as [1], [2], which we directly compare against, and [3].
> We acknowledge that the evaluation of architecture configurations in the main part is not very broad. See Appendix C.1 for experiments evaluating the effect of varying the number of neurons and parameters of the training algorithm. Furthermore, we are conducting additional experiments for sampling deeper networks. The results will be added here in the next few days, once available.
>
> **Regarding learnable parameters:** the trainable parameters and their dimensions are stated at the end of Definition 3.1 on page 3. The full details regarding the setting of the experiments and the architecture configuration are provided in Appendix B.
>
> **Regarding spike-based metrics:** The aim of our paper is to introduce a training algorithm rather than a new architecture or implementation. SynOps and NeuOps measure the inference-time efficiency of a specific architecture and implementation, which is independent of the training method.
>
> **References:**
> [1] Changze Lv, Yansen Wang, Dongqi Han, Yifei Shen, Xiaoqing Zheng, Xuanjing Huang, and Dongsheng Li. Toward relative positional encoding in spiking transformers, 2025.
> [2] Changze Lv, Dongqi Han, Yansen Wang, Xiaoqing Zheng, Xuanjing Huang, and Dongsheng Li. Advancing spiking neural networks for sequential modeling with central pattern generators, 2024.
> [3] Shibo Feng, Wanjin Feng, Xingyu Gao, Peilin Zhao, and Zhiqi Shen. Ts-lif: A temporal segment spiking neuron network for time series forecasting, 2025.

---

> ### Author Response · Authors · 2025-11-19
> **Regarding Weakness 4**
>
> This is a great question aiming at the heart of how S-SWIM differs from other training schemes.
> The cited methods are all based on the iterative minimisation of a global error function through repeated weight updates. These methods differ in how the updates are computed and propagated to hidden layers through the spiking nonlinearity.
> S-SWIM does not perform iterative updates at all. S-SWIM only performs a single pass over the network by directly solving local surrogate objectives for each hidden layer and then fits the output layer by regression. S-SWIM is a random feature method: the weights in S-SWIM are not computed iteratively but rather sampled from a probability distribution, which in our case (similar to SWIM) is constructed from the training data.
> The pipeline described in Algorithm 1 on page 5 is executed as stated. There is no outer optimisation loop.

---

> ### Author Response · Authors · 2025-11-19
> **Adressing Weakness 5**
>
> **Regarding the claims made in the conclusion:** They can all be mapped to evidence or design choices explained in the main part of the paper.
>
> *“Our approach addresses key challenges in training SNNs by circumventing the approximation of the spike function gradient entirely, through the use of the RFM paradigm.”*
>
> The spiking nonlinearity naturally introduces difficulties for gradient-based training of SNNs, as the non-differentiability and sparsity demand approximation schemes. S-SWIM does not propagate errors through spike functions and therefore does not require approximation schemes such as surrogate gradients. Also see the remark regarding Weakness 4.
>
> *“Through numerical experiments, we show that S-SWIM can serve as a solid standalone strategy[…]”*
>
> See the rows SSWIM (Morlet) and SSWIM (Hat) in Table 1 on page 9.
>
> *“[…] or initialisation for gradient-based training, […]”*
>
> See the rows SSWIM+SGD (Morlet) and SSWIM+SGD (Hat) in Table 1 on page 9. Note that those experiments were run with fewer epochs than full SGD training and still achieved comparable or better results in many cases. The relevant details of the experimental settings are found in Appendix B.4.
>
> *“[…] especially at long prediction horizons,[…]”*
>
> Note how S-SWIM outperforms SGD at long horizons on 3 out of 4 datasets and performs more stably across prediction horizons than SGD.
>
> *“[…] while being orders of magnitude faster than full SGD optimisation.”*
>
> See Figure 1 on page 9. Note that the times are plotted on log-scale.
>
> *“Furthermore, the approach is interpretable and modular, easing future theoretical analysis, practical improvements and adaptation to different tasks.”*
>
> Regarding interpretability, see e.g. Section 3.2.3 Sampled Weights: Since the weights are specified as the closed form solution of simple problems (namely the separation of specifically chosen pairs of input samples), S-SWIM sampled networks lend themselves to an almost mechanistic interpretability.
>
> Regarding adaptability, see e.g. Section 3.2.2 Sampling Distribution and Appendix A.1: Which samples should be separated by the network is defined by the choice of distance functions. These can be chosen to incorporate available knowledge about which features in the inputs are relevant to the task. Some examples for this are provided in Appendix A.1 on page 16.
>
> Regarding modularity, see e.g. Section 3.2.4 Normalisation: Since the weights are chosen as normalised eigenvectors, their scale can be chosen independently from the direction, e.g. to achieve desired spiking statistics.
>
> **Regarding credit assignment:** We did not claim that S-SWIM performs better credit assignment. S-SWIM does not perform credit assignment of the global error to the weights of the hidden layers at all. Different to our method, SGD does perform credit assignment. This gets harder for longer prediction horizons. Since S-SWIM doesn’t perform credit assignment, it doesn’t suffer from the problem getting harder. We posit this as a possible explanation for the observation that the performance of networks trained using S-SWIM is more stable across prediction horizons than that of networks trained using SGD.

---

> ### Author Response · Authors · 2025-11-19
> **Adressing Weakness 3**
>
> This is indeed a well-founded criticism and we will improve this in the paper. We will point out the revisions once this is done and the new PDF is uploaded.

---

### Official Review · Reviewer_SPAj · 2025-11-12

**Soundness:** 2
**Presentation:** 1
**Contribution:** 3
**Rating:** 6
**Confidence:** 2

**Summary:**

This paper proposes a gradient-free training method for spiking neural networks (SNNs) based on the spike response model (SRM), leveraging the random feature method. It introduces a sampling strategy for data points, a weight design scheme for temporal parameters, and an approach for handling output delays.

**Strengths:**

**S1.** This paper addresses an important topic — exploring how to train spiking neural networks (SNNs) without relying on gradient-based methods.

**S2.** The incorporation of the random feature method into SNN training is novel, and the experimental results demonstrate its effectiveness to a certain extent.

**Weaknesses:**

**W1.** The paper introduces a large number of notations, which makes it somewhat difficult to follow. In particular, Definition 3.1 employs multiple types of brackets—{}, () and []—without clear distinction (e.g.,$\phi{x}, f(t), and \zeta[v]$), which can be confusing. Moreover, the use of the symbol “!=” should be clarified. Since I am an emergency reviewer with limited time to thoroughly read the paper, this issue significantly affects the readability and comprehension of the technical content.

**Questions:**

**Q1.** According to Figure 1, why do S-SWIM-trained networks fine-tuned with SGD (i.e., SGD-1) require less training time than the S-SWIM-trained networks themselves in some cases?

**Q2.** What is the architecture of the network trained using S-SWIM? Since S-SWIM models each neuron individually, the training time may increase linearly with the number of neurons. Could the authors clarify this relationship?

**Q3.** How sensitive is S-SWIM to the number of random features?

---

> ### Author Response · Authors · 2025-11-19
> **Adressing Weakness 1**
>
> That’s a good catch. The use of the brackets is indeed not fully consistent currently, which we will work over during the revision. We will streamline and revise the use of brackets as follows:
> * Braces “{ }” are only used to denote sets.
> * Square brackets "[ ]" are used for convolutions “[ f * g ]”.
> * Operators now take function and "simple" arguments together: $\mathcal{G}(f, t)$
>
> We will also streamline the notation employed throughout the paper. We will point out the revisions once this is done.
> The “!=” symbol was meant to signify that the following simplification holds given the additional assumptions, namely that the second argument to the computed convolution is always a spike train, which allows application of the sifting property. Since this seems to rather hinder than help understanding, we will remove it to not further increase the employed notation.
>
> **Edit:** Notation convention updated from original comment.

---

> ### Author Response · Authors · 2025-11-19
> **Adressing Question 1**
>
> “SGD 1 Epoch” in Figure 1 is the first epoch of gradient training only. This was included to give an estimate of how many epochs of SGD training S-SWIM roughly corresponds to in terms of runtime.
> The time for the fine-tuning experiments is not reported in the figure.

---

> ### Author Response · Authors · 2025-11-19
> **Adressing Question 2**
>
> **Regarding the architecture:** Appendix B gives the full experimental settings and details about the datasets. Appendix B.4 contains all hyperparameters, including the specific architecture and kernels.
>
> **Regarding the training time**, that is an excellent question. We answer separately, for (1) the solution of the linear problem for the output layer and (2) the construction of the hidden layer weights. In the notation of the paper, let $N_l$ be the number of neurons in layer $l$, and $L+1$ the index of the final layer of a network with $L$ hidden layers. For the hidden layers, each neuron requires finding one eigenvector, so the number of eigenproblems to be solved scales linearly in the number of neurons in the current layer $N_l$. The dimension of each eigenproblem is the number of neurons in the previous layer $N_{l-1}$. For a full factorisation, this would mean cubic scaling in $N_{l-1}$. This is a coarse bound, as we are only interested in a single eigenvector, which can be computed more efficiently. Similarly, the final layer requires the solution of one $(N_{L}+1)\times (N_{L}+1)$ system per neuron in the output layer, so we get the same scaling in the current and previous layers’ neurons for the solution of the linear problems as for solving the eigenproblems. The main observed runtime cost was due to the assembly of the system matrices for the linear problems (See equation (10) on page 8). While this scales linearly in $N_{L+1}$ and quadratically in $N_{L}$, performing the assembly over the full training data set will typically involve CPU-GPU data transfers similar to loading batches in SGD. The observed runtime likely doesn’t reflect the above theoretical considerations, since the transfers are much more expensive than the computation on the GPU and thus likely hide the asymptotic trends for typical network sizes.

---

> ### Author Response · Authors · 2025-11-19
> **Adressing Question 3**
>
> The ablation study to answer this can be found in Appendix C.1.
> On the tested datasets and configurations, we have found that between 50 and 200 neurons, the error decreases quite rapidly with increasing number of neurons, but this trend flattens afterwards. So for the given dataset, the number of features significantly affects the performance between 50 and 200 sampled features, but causes smaller changes for larger numbers of features.

---

> > ### Comment · Reviewer_SPAj · 2025-11-20
> > **Comment After the Rebuttal**
> >
> > Thank you for your response. After the rebuttal, I decide to maintain my original rating.

---

### Official Review · Reviewer_WHnL · 2025-11-13

**Soundness:** 3
**Presentation:** 2
**Contribution:** 1
**Rating:** 4
**Confidence:** 4

**Summary:**

The paper proposes S-SWIM, a gradient-free training algorithm for Spike Response Model (SRM) spiking neural networks based on Random Feature Methods (RFMs) and inspired by the SWIM algorithm previously developed for ANNs. The key idea is to construct hidden-layer weights and temporal parameters (delays, kernel widths) in a data-driven way so that pairs of inputs with similar signals but dissimilar targets are separated in function space, and then to solve a linear problem for the output layer. This avoids backpropagating gradients through the non-differentiable spike function and thus avoids surrogate gradients entirely.

Experiments focus on time-series forecasting on four standard datasets (METR-LA, PEMS-BAY, Solar, Electricity) with multiple prediction horizons. The authors compare: (a) S-SWIM alone, (b) surrogate-gradient SGD (Adam), and (c) S-SWIM used as initialization followed by SGD, and report RSE metrics and training times. S-SWIM often matches or outperforms pure SGD on these forecasting tasks and is reported to be one to three orders of magnitude faster in training time.

**Strengths:**

The paper cleanly adapts data-driven random feature sampling from SWIM to a Spike Response Model SNN setting, including temporal parameters (delays, kernel supports) as part of the random feature construction. This is more principled than purely random ELM-style initialisation and fits well with the SNN temporal structure.

S-SWIM provides a surrogate-gradient-free alternative, addressing a real pain point in SNN training (biased surrogates, gradient instability, heavy compute/memory cost). All learning is done via sampling, linear algebra, and correlation analysis over spike trains and kernel responses.

**Weaknesses:**

All experiments are on tabular / numerical time-series datasets (METR-LA, PEMS-BAY, Solar, Electricity) with RSE metrics.

There are no experiments on canonical SNN benchmarks such as neuromorphic image datasets (e.g., DVS variants), frame-based image classification (MNIST/CIFAR-like), audio/speech spiking tasks, or text/event-driven tasks.

Given the broad claims (“general fast training algorithm for SNNs”, “potentially more energy-efficient alternative to ANNs”) and the heavy emphasis on the generality of the SRM parameterization, the fact that only one task family (time-series forecasting) is actually tested makes the empirical validation feel incomplete. A reader cannot tell whether S-SWIM meaningfully helps in other common SNN application domains such as vision or speech.

The paper focuses on a relatively specific shallow feed-forward architecture (the “shallow network case”) and does not empirically validate S-SWIM on deeper networks, even though the method is claimed to be applicable to them.

There is no evaluation on classification tasks or tasks with spike-valued outputs, although the theory discusses them. This accentuates the feeling that the method is only demonstrated in the easiest setting for S-SWIM: continuous regression with L2 loss.

**Questions:**

Overall, this is a technically solid and well-written paper with a genuinely interesting idea: bringing modern data-driven random-feature sampling into the SNN world to obtain a fully gradient-free training scheme with good training-time savings on time-series forecasting benchmarks. The mathematical treatment and modular algorithm design are clear strengths.

However, from a conference-acceptance point of view, the experimental validation feels too narrow. The method is only demonstrated on numerical time-series forecasting, with no experiments on images, videos, text, or speech, and no demonstration on classification or neuromorphic benchmarks. Given the broad claims and ambitious framing, this limited evaluation undermines the impact.

---

> ### Author Response · Authors · 2025-11-19
> **Adressing Question 1**
>
> This is indeed a well-founded criticism, and we will address this by providing further experimental evaluation, especially regarding the deeper networks case.
> Time-series forecasting was chosen as it aligns well with the spatio-temporal nature of SNNs. As the evaluated benchmarks do not show how to apply S-SWIM to the case of spiking inputs and targets, the suggested benchmarks indeed provide valuable additions to the experimental evaluation.
> In Appendix A.4, we discuss how to use S-SWIM for spike-valued and function-valued targets. While it is currently unclear how to do this for the general case, we will show a proof-of-concept for this idea on a classification problem, since this does not require fitting exact spike times.
> We will point out the additions once the results are available.

---

### Author Response · Authors · 2025-12-03
**Summary for AC**

Dear AC and Reviewers,

We sincerely appreciate the time and effort you have dedicated to reviewing our manuscript. We fully understand the additional workload and challenges brought about by the current situation, and we truly appreciate your continued efforts.

In the hope of facilitating your decision-making process, we would like to provide a summary of the discussion phase:

The main weaknesses raised during the review phase were the narrow experimental evaluation and poor readability.
* Regarding experimental evaluation: The pre-review version included experiments on time series forecasting with explicitly prescribed target voltages and real function-valued inputs. Reviewer WHnL pointed out that networks with multiple hidden layers and datasets with targets that can not directly be regressed against to solve for the output layer weights were missing. We have added a comprehensive suite of tests on classification datasets, which shows how to apply the algorithm to cases where the output of the final layer is in a different form than the targets. Furthermore, we also evaluated deeper networks, showing that the algorithm can be used, but suffers from some performance degradation (as is typical for random feature methods), highlighting avenues for future work. Additionally, this shows that the algorithm can also handle spiking inputs, which was previously only discussed theoretically, but not evaluated numerically, both for real spiking data such as the Spiking Heidelberg Digits dataset and artificial spike trains generated from static images.
* Regarding presentation: Reviewers SPAj, w4Dg, and k7CL pointed out that the work is technically solid and constitutes a novel contribution, but suffers from being very difficult to understand. To this end, they have provided much helpful advice on how to improve the readability of the paper:
   - Reviewer SPAj pointed out that the definition of the network architecture (Definition 3.1) is hard to understand and uses many - partly inconsistent - symbols. To address this, the definition has been reworked thoroughly with a streamlined notation, including references discussing the employed neuron model in more detail. Furthermore, an illustration of the network architecture and spiking neuron model has been added to the paper to ease understanding.
   - Reviewer k7CL suggested adding a figure to illustrate the discussed content. We have added a figure showing both the architecture and model, as well as the main idea of the training algorithm using a simple example.
   - Furthermore, Reviewer k7CL pointed out that the pre-review version discusses the technical details before properly introducing the problem and the intuition of the approach. Taking this suggestion, we have added a summary of the problem we aim to solve at the end of the introduction (see blue revision) and an intuitive discussion preceding the mathematical details at the beginning of the main technical discussion in Section 3.2 (see blue revision).
* Further questions have been raised towards certain referenced details being difficult to find. To this end, table descriptions and free text discussion of the main results have been updated (see blue revisions in section 4).

All revisions and additional clarifications provided during the discussion have been carefully incorporated into the updated manuscript, with changes highlighted in blue for your convenience. We hope this summary can assist your re-evaluation process.

---

### Meta-Review · Area_Chair_Wi7Z · 2025-12-31

**Summary:**

The paper proposes S-SWIM, a data-driven random feature algorithm for training Spiking Neural Networks (SNNs) based on the Spike Response Model (SRM). The authors aim to address the non-differentiability of SNNs by avoiding gradient propagation entirely, instead using a sampling-based approach for hidden layers and a linear solver for the output layer. The submission highlights speed advantages over SGD and potential interpretability.

While the reviewers acknowledged the novelty of adapting the SWIM algorithm to the SNN domain and the impressive training speedups, the consensus leans toward rejection due to significant issues with presentation and experimental breadth. The primary concerns driving this decision are the paper's poor readability and heavy notation, which made the core intuition difficult for multiple reviewers to grasp. Furthermore, the experimental validation was initially restricted to time-series forecasting. While the authors added classification tasks during the rebuttal, the method's performance on deeper networks demonstrated degradation, confirming concerns about the method's scalability and generalizability compared to standard gradient-based approaches.

**Reviewer Concerns:**

### **Addressed Concerns**
**Lack of Classification Experiments**: Reviewer WHnL noted the absence of classification tasks or spike-valued outputs. The authors provided new results on MNIST, Fashion-MNIST, and SHD in the appendix, showing the method can function in these domains.

**Clarification of Baselines**: Reviewer k7CL was confused by the lack of citations for baselines in Table 1. The authors clarified that these were sourced from a specific reference (Lv et al., 2025) and corrected the citation visibility.

**Architecture Visualization**: Reviewer k7CL requested a figure to illustrate the method. The authors added a figure showing the architecture and main training idea.

### **Remaining Concerns**
**Performance on Deep Networks**: Reviewer WHnL questioned the method's applicability to deep networks. The authors' additional experiments admitted that performance degrades with depth, which is a known limitation of Random Feature Methods but significantly limits the paper's impact and claims of being a general alternative to ANNs.

**Lack of Neuromorphic Benchmarks**: While standard classification was added, the evaluation on canonical neuromorphic benchmarks (DVS variants beyond SHD) remains limited compared to the broad claims of energy efficiency and general SNN applicability.

**Reviewer Scores:**

The current ratings reflect a consensus leaning towards rejection, with original scores ranging from a low of 2 to a high of 6. Reviewer k7CL assigned a "Reject" rating of 2, and while they acknowledged the work's novelty post-rebuttal, their predicted post-rebuttal score is unlikely to rise above a 3 as they emphasized that the writing remained "really hard to follow". Reviewers WHnL and w4Dg both assigned a score of 4; their scores are predicted to remain unchanged at 4 because the rebuttal experiments confirmed inherent limitations regarding deeper networks and the fundamental presentation issues persist. The most positive evaluation came from Reviewer SPAj, who assigned a rating of 6; however, despite the authors' clarifications, this reviewer explicitly decided to maintain their original rating after the rebuttal.

---

### Decision · Program_Chairs · 2026-01-26

Reject